SPECIAL ISSUE
LIFELONG DEVELOPMENT

# KLF7 orchestrates hippocampal development through neurogenesis and Draxin-mediated neuronal migration

Yitong Liu[1], Wentong Hong[2], Yuyan Zhou[1], Ailing Zhang[3], Pifang Gong[1], Guibo Qi[1], Xuan Song[1], Zhenru Wang[1], Xuanming Shi[3], Congcong Qi[4] and Song Qin[1,*]

## ABSTRACT

The hippocampus, a brain region that is crucial for cognitive learning, memory and emotional regulation, undergoes its primary development during embryonic and early postnatal stages. Krüppel-like factor 7 (KLF7), a transcription factor associated with autism spectrum disorder and intellectual developmental disorders, plays a pivotal role in brain development. In this study, we investigated the role of KLF7 in hippocampal development using conditional knockout mice [*Emx1*-Cre;*Klf7*<sup>Flox(F)/F</sup>]. We found that KLF7 deletion in hippocampal progenitors resulted in significant hippocampal shrinkage, disrupting neurogenesis, neuronal differentiation and migration. KLF7 mutant mice exhibited abnormal neuronal projections, anxiety- and depression-like behaviors, and memory impairments. Transcriptomic profiling identified Draxin, a neural chemorepellent, as a key downstream target of KLF7. Remarkably, overexpression of Draxin rescued dentate gyrus granule cell migration defects in KLF7 mutant mice. These findings demonstrate that KLF7 is essential for proper hippocampal development and function, regulating neuronal migration through Draxin. This study provides mechanistic insights into the neurological deficits associated with KLF7 pathogenic variants and highlights potential therapeutic targets for neurodevelopmental disorders.

KEY WORDS: Krüppel-like factor 7, Hippocampus, Neurogenesis, Neuronal migration, Neuronal projections, Draxin, Mouse

## INTRODUCTION

The hippocampus is a fundamental brain structure that is essential for cognitive learning, spatiotemporal memory and regulation of emotion (Hainmueller and Bartos, 2020; Knight et al., 2022). Its formation predominantly occurs during the embryonic and early postnatal stages, with the sequential development of its complex architecture being crucial for its proper functioning (Pan et al., 2019). Key components of the hippocampus include the cornu ammonis (CA) and the dentate gyrus (DG) (Deshmukh and Knierim, 2012). In rodents, the migration of pyramidal neurons (PNs) in the CA takes place between embryonic day (E)12 and E18 (Angevine, 1965; Caviness and Sidman, 1973; Kitazawa et al., 2014), primarily originating from the hippocampal neuroepithelium (HNE) in the ventricular zone (VZ), with a small portion arising from basal progenitor cells in the subventricular zone (SVZ) (Deshmukh and Knierim, 2012). The granule neurons of the DG are generated from the dentate neuroepithelium (DNE) and the primary germinative matrix (1°ry GM) (Hatami et al., 2018), with their formation occurring between ∼E13 and postnatal day (P)15 (Bayer, 1980; Seki et al., 2014). Notably, ∼80-85% of dentate granule cells are formed during postnatal development (Bátiz et al., 2016; Bayer, 1980).

The precise migration and positioning of newborn neurons are essential for establishing the hippocampal structure and its functional circuits, with these processes being tightly regulated by extracellular cues and intracellular signaling pathways (Cossart and Khazipov, 2022). Disruptions in neuronal migration can lead to the misplacement of hippocampal neurons and dysfunction in associated brain circuits, which are implicated in various neurodevelopmental disorders, including epilepsy, intellectual disabilities, schizophrenia, and autism spectrum disorders (Alexandre et al., 2006; Guerrini, 2005; Wegiel et al., 2010). However, the mechanisms linking these neuronal migration defects to such conditions remain poorly understood.

Krüppel-like factor 7 (KLF7), a member of the Krüppel-like transcription factor family, is a candidate gene for autism spectrum disorder (ASD), associated with deletions at 2q33.3-q34 (Tian et al., 2022b). Patients with KLF7 pathogenic variants often display intellectual developmental disorders alongside psychiatric characteristics (Powis et al., 2018). Studies have shown that *Klf7*-deficient mice similarly exhibit traits associated with neurological conditions such as autism spectrum disorder and intellectual disability (Angevine, 1965; Tian et al., 2022a). In early postnatal mice, KLF7 is highly expressed in the cerebral cortex, hippocampus, nasal epithelium and trigeminal ganglion. Mice with a global knockout of KLF7 die shortly after birth and exhibit a thinner cerebral cortex, corpus callosum dysplasia and abnormal hippocampal morphology (Lei et al., 2005). Recent findings indicate that KLF7 regulates neurogenesis in the developing mouse cerebral cortex and influences the migration of cortical neurons through downstream effectors such as p21 and Rac3 (Hong et al., 2023).

In this study, we investigated the role of KLF7 in brain development by generating KLF7 conditional knockout mice from anterior neural progenitor cells (NPCs). Compared to littermate controls, KLF7 mutant mice exhibited defects in neurogenesis, impaired cell cycle progression, altered differentiation and disrupted neuronal migration in the hippocampus. Furthermore,

[1]Department of Neurology, Shanghai Pudong Hospital and Department of Histoembryology, School of Basic Medical Sciences, State Key Laboratory of Medical Neurobiology and MOE Frontiers Center for Brain Science, Fudan University, Shanghai, 200032, China. [2]Department of Anatomy, School of Basic Medical Sciences, Bengbu Medical University, Bengbu, 233030, China. [3]Department of Biochemistry, School of Basic Medical Sciences, Anhui Medical University, Hefei, 230032, China. [4]Department of Laboratory Animal Science, Fudan University, Shanghai, 200032, China.

*Author for correspondence (sqin@fudan.edu.cn)

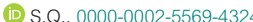 S.Q., 0000-0002-5569-4324

KLF7 mutant mice exhibited abnormal neuronal projections, as well as anxiety- and depression-like behaviors, and impairments in memory function. Through transcriptomic profiling analysis, we identified target genes regulated by KLF7 and demonstrated that KLF7 controls neuronal migration to the DG by regulating the expression of Draxin in the hippocampus.

## RESULTS

### Specific deletion of KLF7 leads to hippocampal shrinkage

Our previous studies have shown that KLF7 is highly expressed in both embryonic and postnatal hippocampus (Hong et al., 2023). To elucidate the role of KLF7 in hippocampal development, we used the *Emx1-Cre* mouse line (Cocas et al., 2009; Gorski et al., 2002) to generate *Emx1-Cre;Klf7*[F/F] conditional knockout (cKO) mice (Fig. S1A). The *Emx1-Cre* transgenic mouse enables genetic recombination in most of neurons of the neocortex and hippocampus, as well as in a limited population of cortical glial cells (Gorski et al., 2002), making it a valuable model for studying forebrain development and function (Hong et al., 2023). Compared to *Klf7*[F/F] controls, the expression of KLF7 was significantly reduced in the postnatal hippocampus of *Emx1-Cre;Klf7*[F/F] mice (Fig. S1B,C).

Nissl staining revealed that at both early postnatal stages (P7 and P14) and in adulthood, but not at the embryonic stage (E18.5) (Fig. 1A), *Klf7* deletion resulted in substantial morphological abnormalities in the hippocampus (Fig. 1B), characterized by a marked reduction in the sizes of the CA and DG regions (Fig. 1C,D). We next examined the PNs and granule neurons in the hippocampus of *Emx1-Cre;Klf7*[F/F] mutants and controls at E18.5, P0 and P7 (Fig. 1E). Consistently, the number of $Ctip2^+$ PNs in CA1-CA2 and granule neurons in the DG was significantly reduced at P0 and P7 in mutants, but not at E18.5 (Fig. 1F,H). $Math2^+$ PNs in CA3 were dramatically decreased at P7, but not at E18.5 or P0 (Fig. 1G). Moreover, the number of $NeuroD^+$ neuronal precursors in the developing postnatal DG region was significantly lower in KLF7 mutants compared to *Klf7*[F/F] controls at P7 (Fig. 1I,J). Together, these data suggest that KLF7 expression in hippocampal NPCs is crucial for the proper development of the hippocampus.

### KLF7 mutant mice exhibit abnormal distribution of NPCs and intermediate progenitor cells in the developing hippocampus

Given that cKO of KLF7 in early hippocampal progenitors dramatically disrupts normal hippocampal development, we next sought to determine whether *Klf7* depletion affects the distribution of NPCs and intermediate progenitor cells (IPCs) in the hippocampus. Immunofluorescence analysis revealed no significant differences in the distribution of hippocampal NPCs ($PAX6^+$) and IPCs ($TBR2^+$) between *Emx1-Cre;Klf7*[F/F] mutants and *Klf7*[F/F] control mice at E14.5 (Fig. 2A-C,E,F).

To assess the distribution of proliferating progenitor cells at E14.5, we performed intraperitoneal injections of EdU in pregnant mice at E14.5. The results showed that the proliferation of all progenitor cells at this stage remained within normal limits (Fig. 2D,G).

We next examined the distribution of hippocampal NPCs ($PAX6^+$) and IPCs ($TBR2^+$) in *Emx1-Cre;Klf7*[F/F] mutants compared to *Klf7*[F/F] control mice at E18.5. Based on the structural characteristics of hippocampal development at E18.5 and the distribution of $PAX6^+$ NPCs, the developing hippocampus was divided into four distinct regions for individual analysis: HNE; 1ry GM; the secondary germinative matrix (2ry GM); and the tertiary germinative matrix (3ry GM) (Fig. 2H,I). For the distribution of $TBR2^+$ IPCs, the hippocampus was segmented into three regions: HNE; the dentate migratory stream (DMS; which includes the 1ry GM and 2ry GM); and 3ry GM (Fig. 2K,L). Immunofluorescence analysis revealed a significant reduction in the distribution of NPCs ($PAX6^+$) in HNE, 1ry GM and 2ry GM, but not in 3ry GM, in *Emx1-Cre;Klf7*[F/F] mice at E18.5 (Fig. 2J). Similarly, there was a marked reduction in IPCs ($TBR2^+$) in the HNE and DMS regions, but no significant change in the 3ry GM in *Emx1-Cre;Klf7*[F/F] mice at E18.5 (Fig. 2M).

The above results suggest that cKO of KLF7 leads to a significant reduction in NPCs and IPCs in the HNE and DMS regions, but not in the 3ry GM at E18.5. However, postnatal analysis revealed abnormalities in both $Ctip2^+$ granule neurons and $NeuroD^+$ neuronal precursor cells in the DG (Fig. 1H,J). To further investigate this issue, we performed immunohistological analysis of TBR2 expression in the postnatal developing hippocampus. At P0, based on the structural characteristics of hippocampal development and the distribution of $TBR2^+$ IPCs, we segmented the developing hippocampus into two distinct regions: DMS and 3ry GM (Fig. 3A,B). We found that the number of $Tbr2^+$ IPCs in the DMS and 3ry GM was significantly reduced in *Emx1-Cre;Klf7*[F/F] mutant mice compared to *Klf7*[F/F] controls at P0 (Fig. 3C). We then monitored the distribution of $TBR2^+$ IPCs at different postnatal time points (P3, P7 and P14) in the developing DG (Fig. 3D,G,J). At P3 and P7, there was a dramatic reduction in the number of $Tbr2^+$ IPCs in KLF7 mutant mice compared to controls (Fig. 3E,F,H,I). At P14, the expression of $TBR2^+$ IPCs in *Klf7*[F/F] controls was confined to the subgranular zone (SGZ) of the DG, whereas in *Emx1-Cre;Klf7*[F/F] mice, $TBR2^+$ IPCs were aberrantly distributed throughout the hilus (Fig. 3K,L). Together, these data demonstrate that the specific deletion of *Klf7* leads to an abnormal distribution of NPCs and IPCs in the embryonic hippocampus, particularly in the HNE and DMS regions, and a disrupted distribution of IPCs in the postnatal DG.

### KLF7 regulates the cell cycle progression and differentiation of hippocampal NSCs

To investigate the underlying cause of the reduced NPC and IPC populations in the KLF7 cKO hippocampus, we first evaluated apoptotic activity during development. Immunostaining for cleaved caspase-3 revealed no significant difference in programmed cell death between *Klf7*-deficient mice and *Klf7*[F/F] controls at E14.5. Similarly, cleaved caspase-3$^+$ cells remained unchanged at E17.5, P0 and P7 (Fig. S2A,B). We then assessed cell cycle dynamics during embryonic neurogenesis. Between E12 and E18, NPCs display a short cell cycle (~8-18 h) and undergo robust proliferation (Takahashi et al., 1993, 1995). To track proliferative activity, EdU was administered, and embryos were collected 24 h later. Proliferating progenitor cells were then identified by PCNA staining. Cells that were labeled with EdU but not PCNA were considered progenitors that had exited the cell cycle. We first examined the HNE region. At E15.5, *Klf7*-deficient mice showed a decreased tendency for progenitor cells to exit the cell cycle, while no significant differences were observed at other time points (Fig. 4A,C). To assess whether the reduced number of progenitors exiting the cell cycle was linked to changes in the numbers and distribution of NPCs and IPCs, we co-labeled for EdU, PAX6 and TBR2 in the HNE region (Fig. 4B). At E15.5 and E16.5, *Emx1-Cre;Klf7*[F/F] mutant mice exhibited a significantly higher number of NPCs ($EdU^+PAX6^+/EdU^+$), whereas *Klf7*[F/F] controls showed fewer IPCs ($EdU^+TBR2^+/EdU^+$) (Fig. 4D,E). These findings suggest that

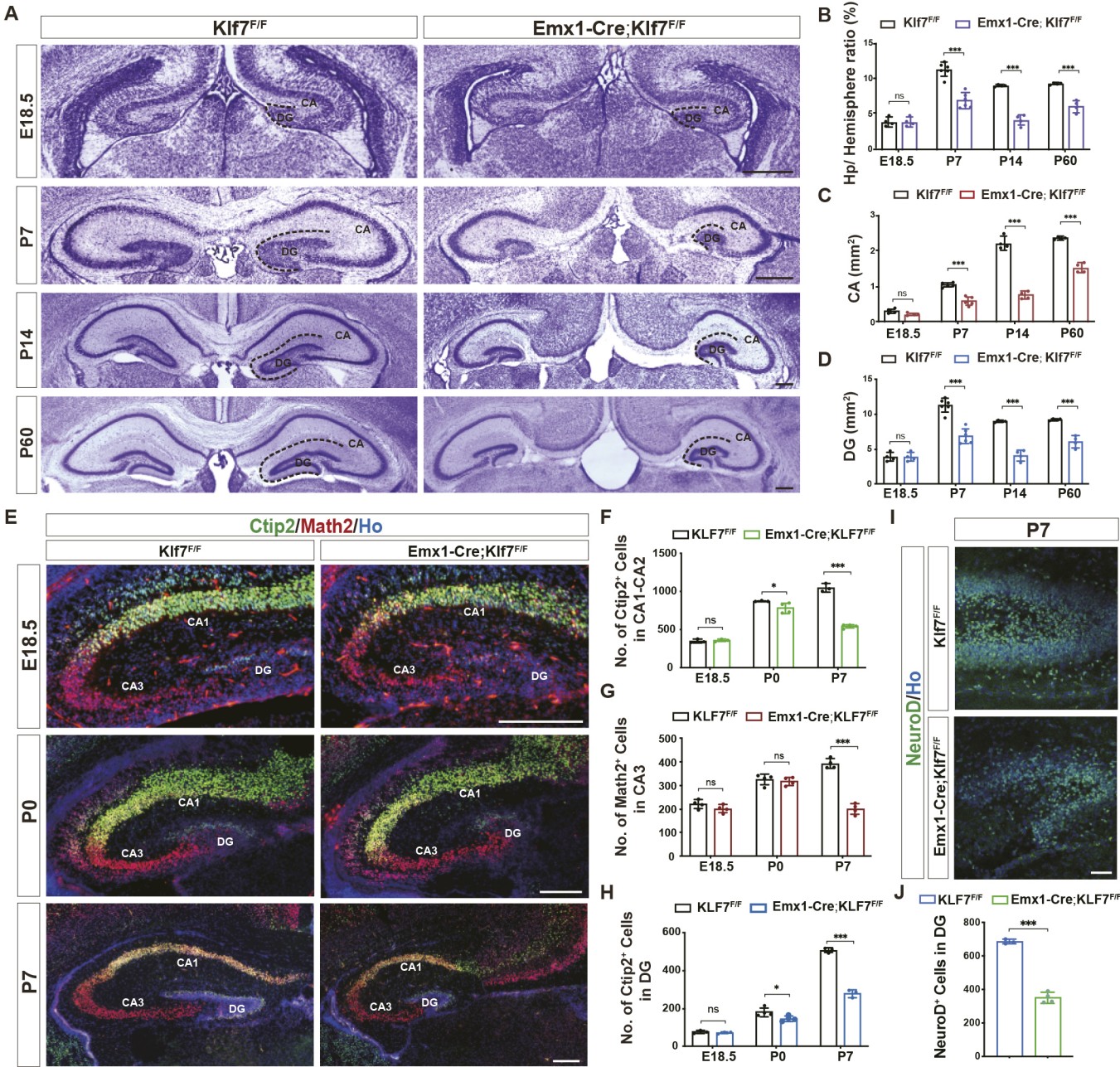

**Fig. 1. KLF7 deletion disrupts hippocampal development.** (A) Nissl staining of hippocampus at different stages. Dashed lines indicate CA-DG interface. (B-D) Quantification of hippocampal, CA, and DG areas in $Klf7^{F/F}$ and $Emx1\text{-}Cre;Klf7^{F/F}$ mice ($n$=4-6). (E) Immunofluorescence for Ctip2 (green) and Math2 (red) in hippocampus at E18.5, P0 and P7. (F-H) Quantification of Ctip2$^+$ (CA1-CA2, DG) and Math2$^+$ (CA3) cells ($n$=3-4). (I,J) NeuroD immunofluorescence and quantification in DG at P7 ($n$=3-4). ns, not significant; *$P$<0.05; ***$P$<0.001 (unpaired two-tailed $t$-test). Data are mean±s.e.m. CA, cornu ammonis; DG, dentate gyrus. Scale bars: 500 µm (A); 200 µm (E); 50 µm (I).

$Klf7$ deficiency specifically delays the exit of progenitor cells from the cell cycle at E15.5 in HNE region, thereby impairing the differentiation of NPCs into IPCs.

Next, we examined the 1ry GM and the 2ry GM regions. In the 1ry GM, $Klf7$ deficiency was associated with an increased cell cycle length at E15.5 (Fig. 4F,H), whereas no significant differences were observed in the 2ry GM (Fig. S2C). To investigate which progenitor subtypes were most affected by $Klf7$ ablation, we stained for PAX6 for NPCs and TBR2 for IPCs 1 day after EdU labeling (Fig. 4G). We observed a higher number of EdU$^+$PAX6$^+$TBR2$^+$/EdU$^+$ cells in $Klf7$ cKO embryos at E15.5 and E18.5 (Fig. 4I), suggesting an

accumulation of progenitor cells in a transitional state between NPCs and IPCs. Moreover, there were significantly fewer IPCs (EdU$^+$PAX6$^-$TBR2$^+$/EdU$^+$) in $Klf7$-deficient mice at E15.5 (Fig. 4J). These results indicate that $Klf7$ deficiency specifically delays progenitor cell cycle exit in the 1ry GM at E15.5, leading to an increased number of progenitor cells in the transitional state and delaying their differentiation into IPCs in both the 1ry and 2ry GMs.

Finally, we examined the 3ry GM region. At E16.5 and E18.5, $Klf7$-deficient mice exhibited a decreased tendency to exit the cell cycle (Fig. 4K,M). One day after EdU labeling, we performed PAX6 and TBR2 staining (Fig. 4L) and found a higher number of

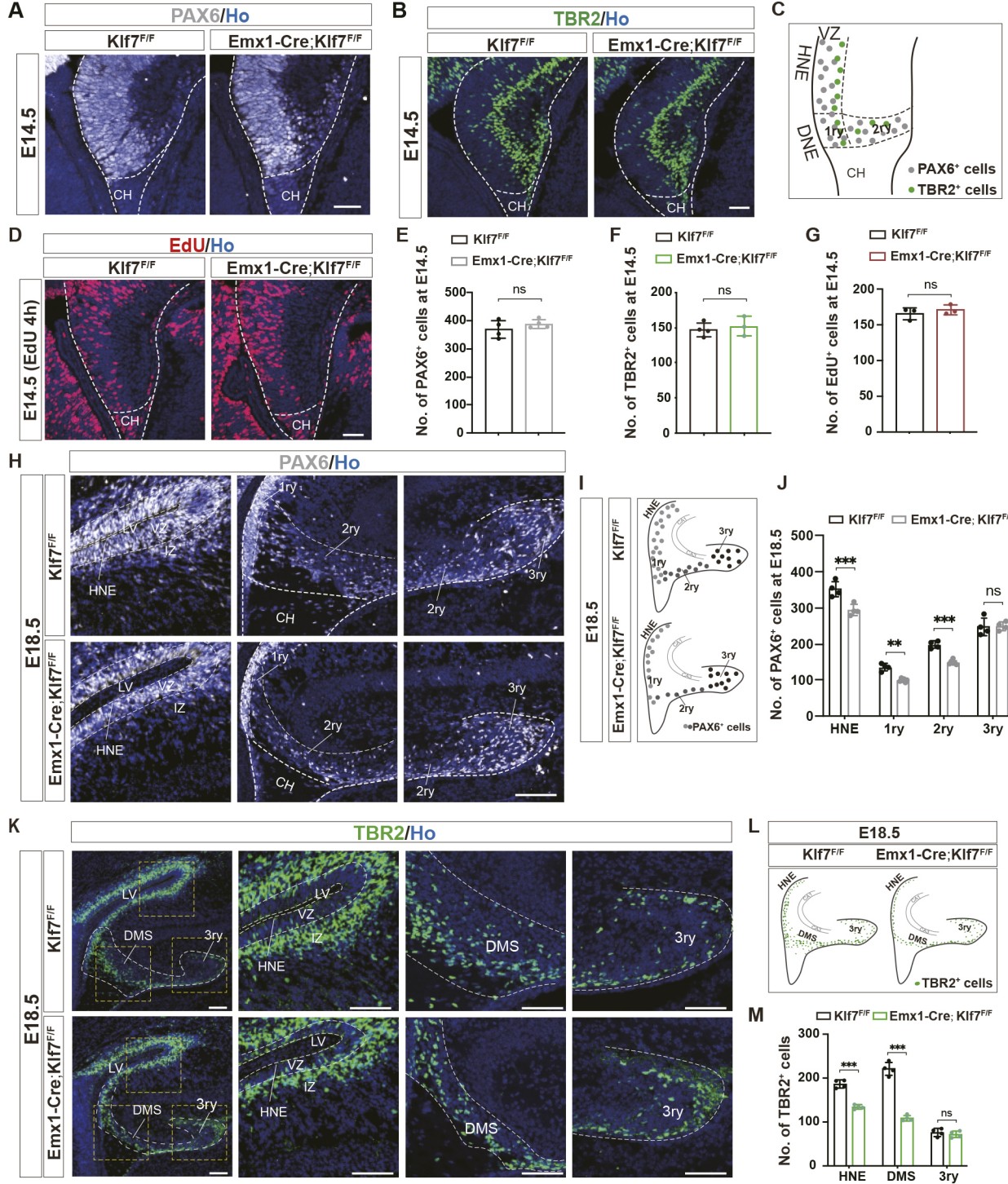

**Fig. 2. KLF7 cKO reduces hippocampal NPCs and IPCs.** (A,B) PAX6 and TBR2 immunostaining at E14.5. Dashed lines indicate hippocampal region. (C) Schematic distribution of PAX6+ and TBR2+ cells at E14.5. (D) EdU staining at E14.5. Dashed lines indicate hippocampus. (E-G) Quantification of PAX6+, TBR2+ and EdU+ cells at E14.5 (*n*=3-4). (H-M) PAX6 and TBR2 immunostaining, schematic distribution and quantification at E18.5 (*n*=4). Dashed lines indicate the distinct developing regions of the hippocampus. ns, not significant; **$P<0.01$; ***$P<0.001$ (unpaired two-tailed *t*-test). Data are mean±s.e.m. CH, cortical hem; DMS, dentate migratory stream; DNE, dentate neuroepithelia; HNE, hippocampal neuroepithelium; IZ, intermediate zone; LV, lateral ventricles; VZ, ventricular zone; 1ry, primary germinative matrix; 2ry, secondary germinative matrix; 3ry, tertiary germinative matrix. Scale bars: 500 μm (A,B,D); 200 μm (H,K).

EdU+PAX6+TBR2+/EdU+ cells in *Klf7* cKO embryos at E16.5 and E18.5 (Fig. 4N). Additionally, there was a significant reduction in the number of IPCs (EdU+PAX6−TBR2+/EdU+) in *Emx1-Cre; Klf7*F/F mutant mice at E18.5 (Fig. 4O). These findings indicate that *Klf7* deficiency specifically delays progenitor cell cycle exit at

E16.5 and E18.5 in the 3ry GM, which in turn impacts the differentiation of NPCs into IPCs. Taken together, our results highlight the crucial role of KLF7 in regulating the cell cycle progression of NPCs and their differentiation dynamics within the hippocampus.

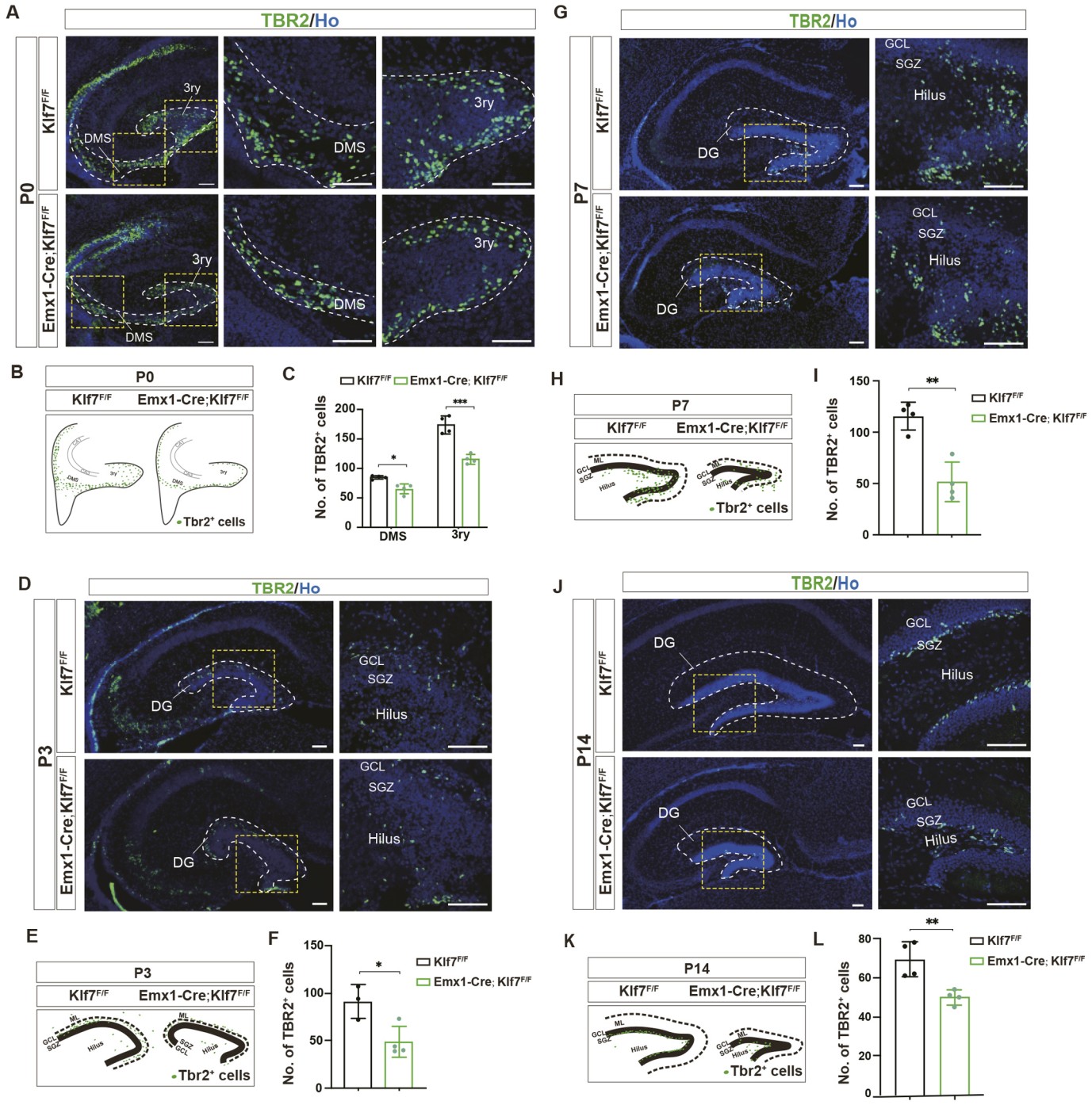

**Fig. 3. KLF7 cKO reduces postnatal DG neural precursor cells.** (A-L) TBR2 immunostaining (A,D,G,J), schematic distribution (B,E,H,K) and quantification (C,F,I,L) in hippocampus at P0, P3, P7 and P14 (*n*=3-4). ns, not significant; *P<0.05; **P<0.01; ***P<0.001 (unpaired two-tailed *t*-test). Data are mean±s.e.m. DG, dentate gyrus; DMS, dentate migratory stream; GCL, granule cell layer; ML, molecular layer; SGZ, subventricular zone; 3ry, tertiary germinative matrix. Scale bars: 100 μm.

## Deletion of KLF7 impairs neuronal migration during hippocampal development

To investigate the impact of KLF7 deletion on neuronal migration during hippocampal development, we intraperitoneally injected EdU into pregnant mice at E12.5 and E14.5, and analyzed samples from their offspring at P7 (Fig. 5A). Following EdU injection at E12.5, we observed that only a small number of EdU+ cells had migrated to the hippocampus by P7 (Fig. 5B). Quantification of EdU+ cells in the CA and DG regions revealed a significant reduction in the number of

E12.5-labeled cells that migrated to these regions in in *Emx1-Cre; Klf7*^F/F^ mutant mice compared to the *Klf7*^F/F^ control group (Fig. 5C,D). As Emx1 is specifically expressed in the forebrain, we also quantified the number of EdU+ cells in the hypothalamus. As expected, the migration of E12.5-labeled EdU+ cells in the hypothalamus was unaffected in both groups (Fig. 5E). During the peak period of neuronal migration at E14.5, we injected EdU into pregnant mice and observed a substantial number of EdU+ cells migrating to the hippocampus (Fig. 5F). The results showed a significant reduction in the number of

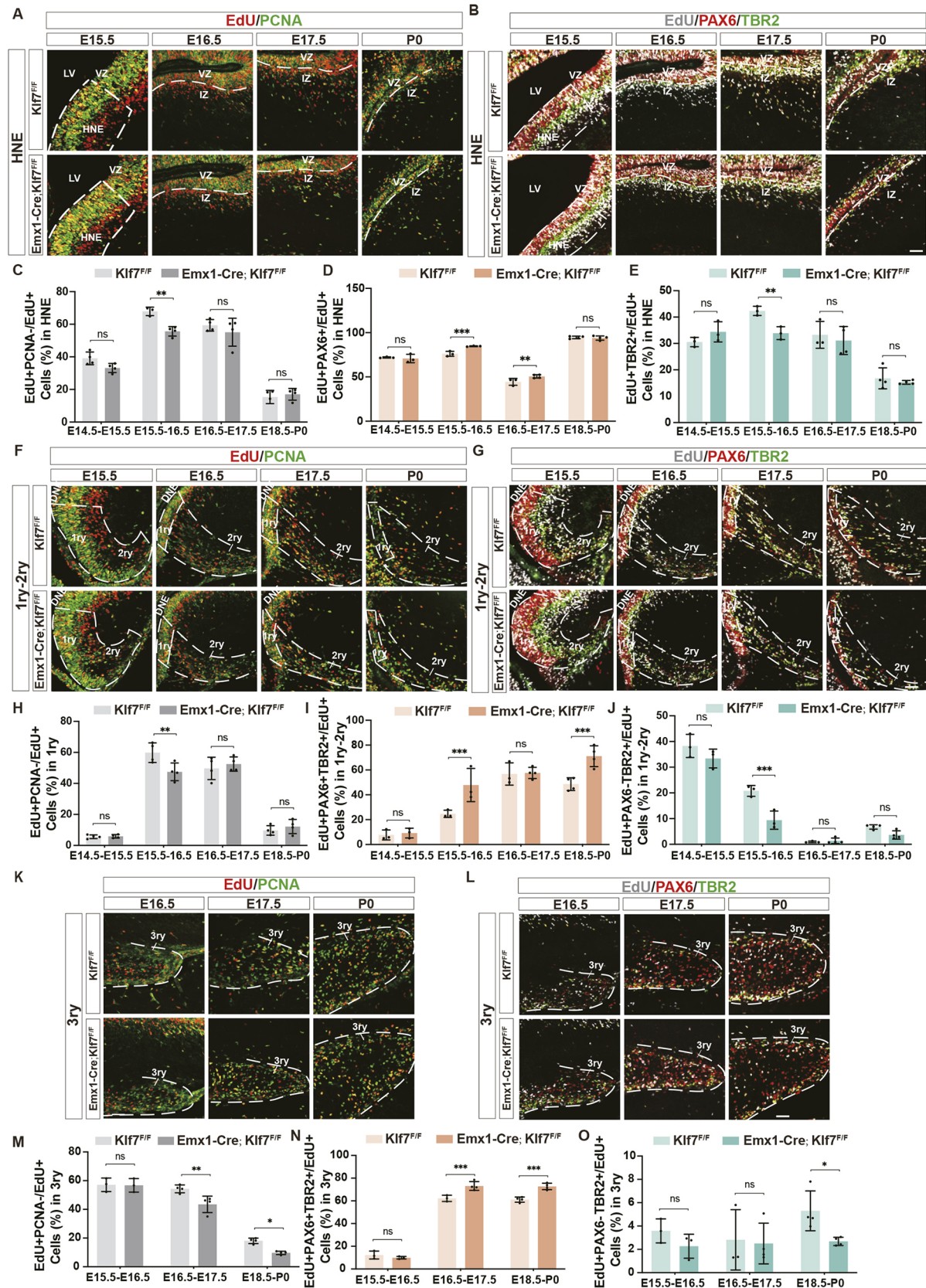

**Fig. 4. KLF7 regulates neural progenitor cell cycle and differentiation.** (A,B) EdU and PCNA, PAX6, TBR2 immunostaining from E15.5 to P0. Dashed lines indicate distinct regions. (C-E) Quantification of EdU⁺PCNA⁻, EdU⁺PAX6⁺ and EdU⁺TBR2⁺ cells (*n*=3-4). (F-O) Similar analysis at different hippocampal subregions and time points. ns, not significant; *P<0.05; **P<0.01; ***P<0.001 (unpaired two-tailed *t*-test). Data are mean±s.e.m. DNE, dentate neuroepithelia; HNE, hippocampal neuroepithelium; IZ, intermediate zone; LV, lateral ventricles; VZ, ventricular zone; 1ry, primary germinative matrix; 2ry, secondary germinative matrix; 3ry, tertiary germinative matrix. Scale bars: 50 µm.

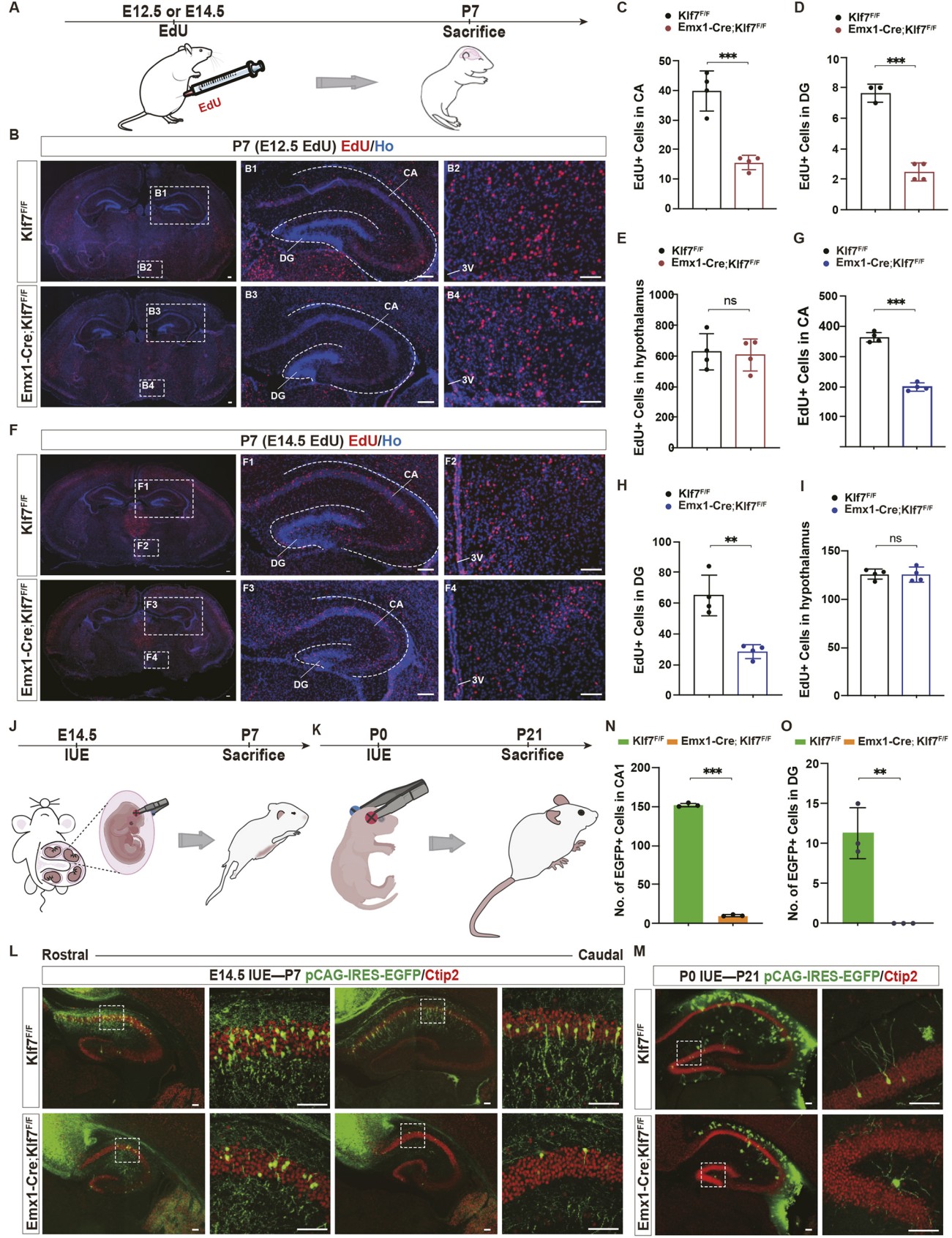

**Fig. 5. KLF7 cKO disrupts neuronal migration.** (A) EdU injection timeline at E12.5/E14.5; analysis at P7. (B-I) EdU immunostaining and quantification in CA, DG and hypothalamus (*n*=3-4). Dashed lines show the interface between the CA and DG. (J-O) Schematic, immunostaining and quantification of *in utero* electroporation (IUE) at E14.5/P0 and neuronal migration analysis at P7/P21 (*n*=3). ns, not significant; ***P*<0.01; ****P*<0.001 (unpaired two-tailed *t*-test). Data are mean±s.e.m. 3V, third ventricle. Scale bars: 100 μm.

E14.5-labeled cells that migrated to the CA and DG regions in *Emx1-Cre;Klf7*^F/F^ mutant mice compared to controls (Fig. 5G,H). However, no significant differences were observed in the hypothalamus (Fig. 5I).

We also conducted *in utero* electroporation (IUE) and postnatal electroporation experiments to examine the migration of neurons to CA and DG regions, respectively. The pCAG-IRES-EGFP expression vector was injected and electroporated into the VZ of mice at E14.5 or P0, and brain tissues were collected at P7 or P21 for analysis using Ctip2 immunofluorescence staining to assess neuronal migration (Fig. 5J,K). At P7 (E14.5 IUE), numerous EGFP⁺ cells were observed along the anterior-posterior (AP) axis of the CA region in the *Klf7*^F/F^ control mice. In contrast, only a few EGFP⁺ cells were found in the coronal plane near the rostral side in the *Emx1-Cre; Klf7*^F/F^ mutant mice (Fig. 5L,N). In P21 mice (electroporated at P0), a substantial number of EGFP⁺ cells were present in the granule cell layer of the DG in the *Klf7*^F/F^ control mice, whereas no EGFP⁺ cells were observed migrating into the DG granule cell layer in *Klf7* mutant mice (Fig. 5M,O). These results suggest that cKO of *Klf7* in NPCs impairs neuronal migration to both the CA and DG regions.

Another notable observation was the mislocalization of both EdU⁻ and EGFP-labeled cells in whole brain sections from the *Emx1-Cre;Klf7*^F/F^ mutant mice compared to controls. At corresponding coronal levels, Klf7 mutant mice showed excessive accumulation of these markers in the retrosplenial granular cortex

compared to controls (Fig. S3A,B). Taken together, these findings emphasize that KLF7 plays a crucial role in regulating neuronal migration during hippocampal development.

### Deletion of KLF7 causes defects in dendritic spines of hippocampal neurons

Immunofluorescence staining at P7 revealed reductions in vGLUT1⁺ presynaptic terminals (Fig. 6A,B) and synaptoporin⁺ mossy fibers (Fig. 6C,D) in the hippocampus, indicating synaptic deficiencies in *Emx1-Cre;Klf7*^F/F^ mutant mice. Golgi staining in adulthood further demonstrated a significant decrease in the anatomical synaptic density (Fig. 6E). Additionally, the number of dendritic spines was notably reduced in both CA1 PNs and DG granule cells (Fig. 6F-I). Taken together, these findings suggest that the specific deletion of *Klf7* leads to a marked reduction in the protrusions of hippocampal excitatory neurons, a decline in dendritic spine density and impaired synaptic formation. This underscores the essential role of KLF7 in the formation and maintenance of hippocampal dendritic structures.

### Deletion of KLF7 leads to impaired axon projection to the dorsal CA1

Neuronal dendrites serve as input sites, receiving signals from other neurons and playing a crucial role in neuronal projections (Fischer

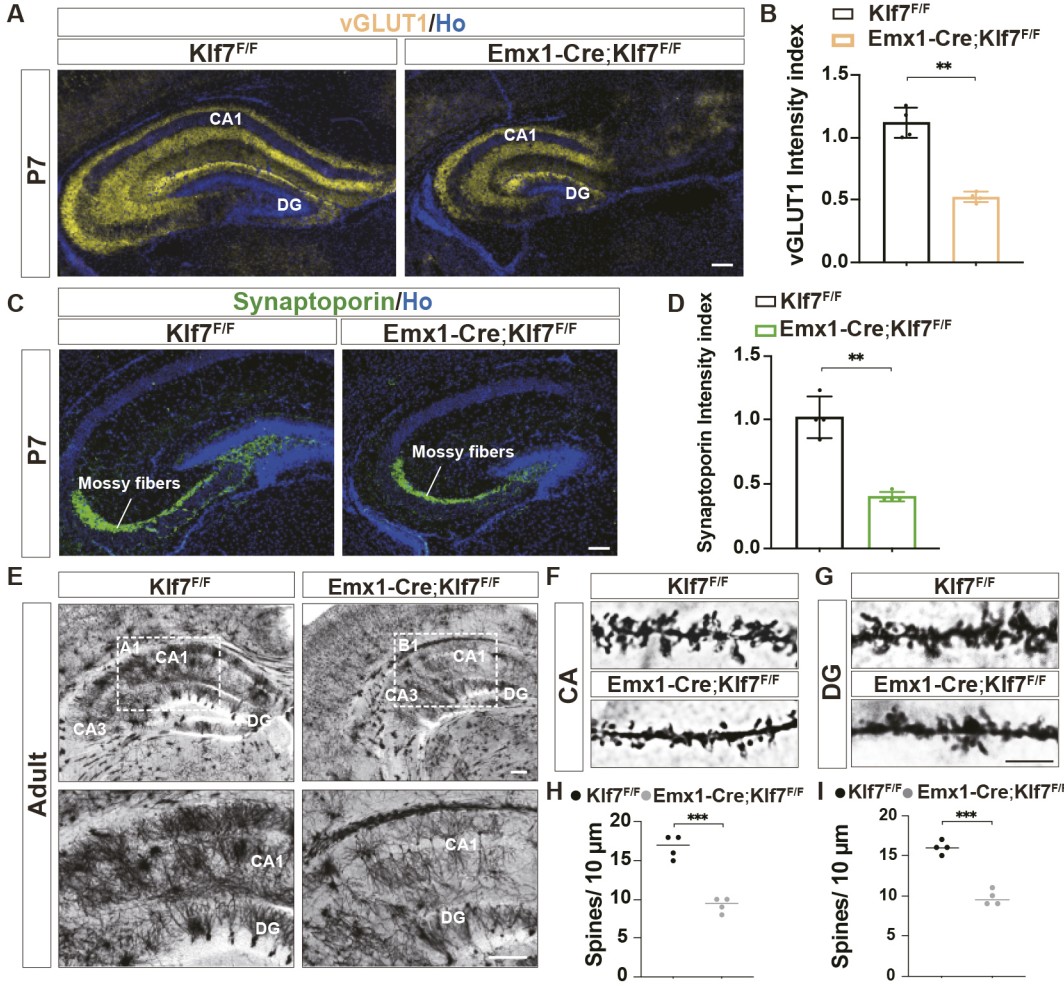

**Fig. 6. KLF7 cKO reduces dendritic spines.** (A-D) Vglut1 and synaptoporin immunostaining and quantification at P7 (*n*=4). (E-I) Golgi staining and dendritic spine quantification in CA and DG (*n*=4). **\**P*<0.01; \*\*\**P*<0.001 (unpaired two-tailed *t*-test). Data are mean±s.e.m. Scale bars: 100 µm (A,C); 200 µm (E), 5 µm (F,G).

et al., 2022). To investigate the impact of *Klf7* deletion on neuronal projections within the dorsal hippocampus, we unilaterally injected a pAAV-hSyn-EGFP-P2A-3XFLAG retrograde trans-monosynaptic virus into the dorsal CA1 (dCA1) of 6-month-old adult mice and systematically analyzed the direct inputs to dCA1 PNs. We found that in *Klf7*^F/F control mice, the majority of inputs to dCA1 PNs originated from the CA regions of both the ipsilateral and contralateral hippocampi (Fig. 7A,B). Across coronal planes along the AP axis, the projection intensity from CA to dCA1 gradually decreased (Fig. 7C,D). In contrast, *Emx1-Cre;Klf7*^F/F mutant mice exhibited dCA1 PNs receiving most monosynaptic inputs from ipsilateral CA and DG regions (Fig. 7E) while completely lacking inputs from the

contralateral hippocampus (Fig. 7F). The projection intensity from ipsilateral CA to dCA1 showed minor variations along the AP axis (Fig. 7G), but near the caudal region, limited GFP fluorescence was observed within the stratum lacunosum-molecular (SLM) layer of the contralateral CA1 (Fig. 7H). These findings indicate that KLF7 deletion significantly disrupts neuronal projections from the contralateral hippocampus to dCA1 and alters the projection patterns from the ipsilateral hippocampus to dCA1.

A small number of neurons from the entorhinal cortex (ENT) and ectorhinal cortex (ECT) project directly to the hippocampal CA1 region via perforant path fibers that traverse the molecular layer and terminate on CA1 dendrites (Tao et al., 2021). Our study

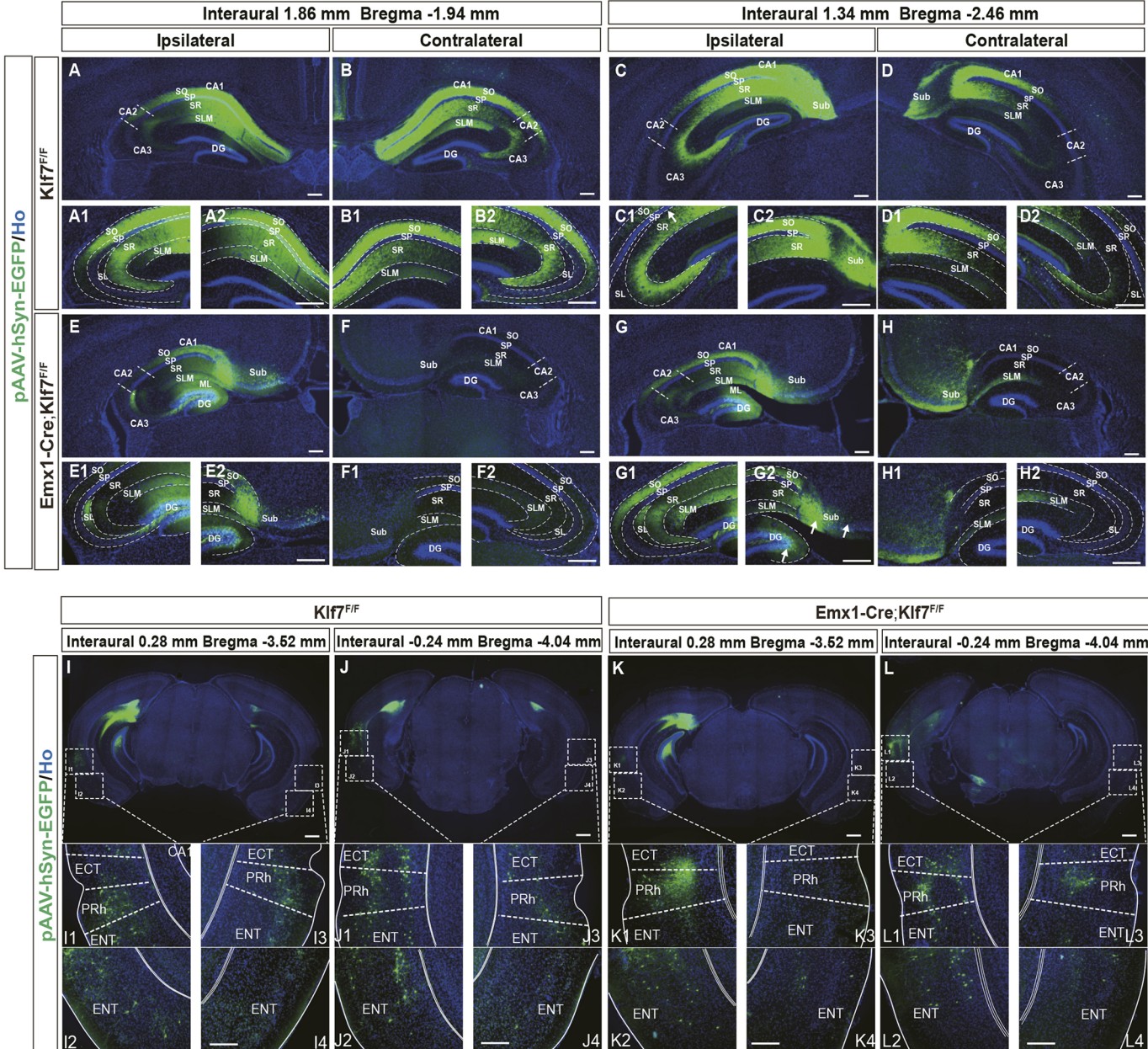

**Fig. 7. KLF7 cKO disrupts dCA1 projections.** (A-H) Coronal sections showing monosynaptic input labeling to ipsilateral and contralateral dCA1 PNs along the RC axis in *Klf7*^F/F and *Emx1-Cre;Klf7*^F/F mice. (I-L) Monosynaptic cortical input labeling to dCA1 in both groups. Dashed lines indicate distinct regions of the hippocampus and cortex. Arrows show various hippocampal subregions, including the boundary between the SP and SR layers, the sub region, and the DG. CA, cornu ammonis; DG, dentate gyrus; ECT, ectorhinal cortex; ENT, entorhinal cortex; ML, molecular layer; PRh, perirhinal cortex; SL, stratum lucidum; SLM, stratum lacunosum-moleculare; SO, stratum oriens; SP, stratum pyramidale; SR, stratum radiatum; Sub, subiculum. Scale bars: 200 μm (A-H); 500 μm (I-L).

consistently demonstrates that in the $Klf7^{F/F}$ control group, dCA1 PNs receive weak direct inputs from the ipsilateral and contralateral ECT, perirhinal cortex (PRh) and ENT (Fig. 7I). Along the AP axis, the projection intensity from these cortical regions to dCA1 exhibits subtle variations (Fig. 7J). In contrast, $Emx1\text{-}Cre;Klf7^{F/F}$ mutant mice showed a complete blockade of projections from the ECT, PRh and ENT to the contralateral dCA1 (Fig. 7K), while projections from these cortical regions to the caudal side remained unaffected (Fig. 7L). These findings indicate that the specific deletion of $Klf7$ significantly disrupts axonal projections within the hippocampus.

### KLF7 cKO mice display depressive- and anxiety-like behaviors as well as memory impairments

Given the essential role of hippocampal connectivity in spatial navigation, memory consolidation, and anxiety- and depression-related behaviors, a series of behavioral tests was conducted to investigate the effects of KLF7 deletion. Firstly, open field tests revealed that $Emx1\text{-}Cre;Klf7^{F/F}$ mice displayed reduced exploratory behavior and heightened anxiety, as evidenced by their reluctance to explore the center regions of open fields (Fig. 8A-C) and increased latency to enter the center zone (Fig. 8D). Notably, the distance traveled and velocity were not significantly altered in these mice (Fig. S4A,B). Secondly, depression-related behaviors were observed in the tail suspension test (TST) and forced swimming test (FST), where $Emx1\text{-}Cre;Klf7^{F/F}$ mice showed significantly prolonged immobility times (Fig. 8E,F). Thirdly, in the elevated plus maze (EPM) test, while the total distance traveled and time spent in open arms were unchanged (Fig. 8G; Fig. S4C), $Emx1\text{-}Cre;Klf7^{F/F}$ mice showed significantly fewer entries into the open arms (Fig. 8H), indicating increased anxiety. Similarly, in the marble burying test (MBT), $Klf7$ cKO mice buried significantly fewer marbles than controls (Fig. 8I,J), highlighting defects in repetitive/stereotypic behaviors.

Memory impairments were also evident. In the Y-maze test, $Emx1\text{-}Cre;Klf7^{F/F}$ mice demonstrated a marked decline in short-term memory, as shown by reduced alternation performance (Fig. 8K) and longer latency to complete the alternation task (Fig. 8L). In the Morris Water Maze test, $Emx1\text{-}Cre;Klf7^{F/F}$ mice took longer to locate the platform during training (Fig. 8M,N) and exhibited fewer platform crossings (Fig. 8O) and reduced time spent in the target quadrant during the probe trial (Fig. 8P). Additionally, the latency to reach the platform zone was significantly increased (Fig. 8Q), although no significant changes in distance traveled or velocity were observed (Fig. S4E,F).

These findings collectively demonstrate that the specific deletion of $Klf7$ results in depressive- and anxiety-like behaviors, along with significant deficits in short-term and spatial memory, underscoring the crucial role of KLF7 in hippocampal function and behavior.

### KLF7 controls neuronal migration to the DG by regulating Draxin

To elucidate the molecular mechanisms underlying hippocampal hypoplasia caused by $Klf7$ mutation, hippocampal tissues from P7 controls and $Emx1\text{-}Cre;Klf7^{F/F}$ mice were subjected to RNA-sequencing (RNA-seq) transcriptome analysis. The data revealed 139 upregulated and 166 downregulated genes in the $Emx1\text{-}Cre;Klf7^{F/F}$ hippocampus (Fig. 9A; Fig. S5A). Gene ontology (GO) analysis of differentially expressed genes (DEGs) highlighted pathways involved in neurogenesis, cell projection organization, mitotic cell cycle transition and dendrite development (Fig. 9B). A subset of DEGs, including those implicated in neuronal

differentiation and migration, was validated using RT-qPCR (Fig. 9C).

Among these, Draxin, a neural chemorepellent critical for axon navigation and neuronal migration during embryonic development (Islam et al., 2009; Shinmyo et al., 2015), was notably downregulated and identified in heat map analysis (Fig. S5B). To investigate the regulatory relationship between KLF7 and Draxin, 293T cells were co-transfected with a KLF7-overexpressing vector and luciferase reporter constructs (pGL4-Basic, pGL4-Draxin or pGL4-Rac3). KLF7 overexpression significantly increased the luciferase activity of pGL4-Draxin, suggesting that KLF7 positively regulates Draxin promoter activity (Fig. 9D). Consistently, in situ hybridization showed reduced Draxin mRNA distribution in the hippocampus of $Emx1\text{-}Cre;Klf7^{F/F}$ mice (Fig. 9E,F). Immunostaining with Draxin and Math2 antibodies further revealed a significant reduction in Draxin$^+$ cells in mutant hippocampi compared to controls (Fig. 9G,H). These results demonstrate that KLF7 deletion causes marked downregulation of Draxin at both mRNA and protein levels.

To confirm the role of Draxin as a downstream effector in KLF7-regulated neuronal migration and preclude non-specific overexpression artifacts, we electroporated precursor cells of the VZ at P0 with either EGFP (2.5 µg/µl; control) or Draxin-EGFP (1.5 µg/µl) plasmids. Hippocampal sections were analyzed at P21 (Fig. 9I). $Emx1\text{-}Cre;Klf7^{F/F}$ brains exhibited significant migration defects of EGFP$^+$ cells in the DG compared to controls. However, Draxin-EGFP overexpression notably enhanced the number of Draxin-EGFP$^+$ cells in the mutant DG (Fig. 9J). To further validate these findings, Draxin-EGFP or EGFP plasmids were electroporated in utero into VZ precursor cells at E14.5, and hippocampal sections were analyzed at P7 (Fig. 9K). While Draxin overexpression failed to rescue migration defects in the CA region of mutant mice (data not shown), Draxin-EGFP$^+$ cells were ectopically located in the DG region of $Klf7^{F/F}$ mice (Fig. 9L). These results demonstrate that Draxin selectively promotes neuronal migration to the DG and plays a crucial role in region-specific hippocampal development.

### DISCUSSION

Hippocampal development is orchestrated by a complex interplay of temporal and spatial signaling cues (Breau et al., 2017; Cossart and Khazipov, 2022; Hatanaka et al., 2016; Khalaf-Nazzal and Francis, 2013). For functional neural circuits to form, neurons must undergo neurogenesis and migrate to their designated locations, but the precise in vivo coordination of these processes remains partially understood. In this study, using neural progenitor-specific KLF7 knockout ($Emx1\text{-}Cre;Klf7^{F/F}$) mice, we demonstrated that the absence of KLF7 disrupts neurogenesis, impairing cell cycle progression, differentiation and neuronal migration in the developing hippocampus. Furthermore, we identified that KLF7 regulates neuronal migration to the DG via its downstream effector Draxin, emphasizing the crucial role of KLF7 in hippocampal development.

KLF7 is highly expressed in the cortex and hippocampus during embryonic and early postnatal stages, with its expression declining in adulthood (Hong et al., 2023; Laub et al., 2001). This temporal expression pattern underscores its essential role in nervous system development. Homozygous KLF7 null mice exhibit high mortality within the first 3 days postnatally, with severely underdeveloped olfactory bulbs (OBs) being the most prominent anatomical abnormality (Laub et al., 2006). Despite high KLF7 expression in the cerebellum, no significant defects have been observed in this region (Laub et al., 2005; Lei et al., 2005), suggesting its specific

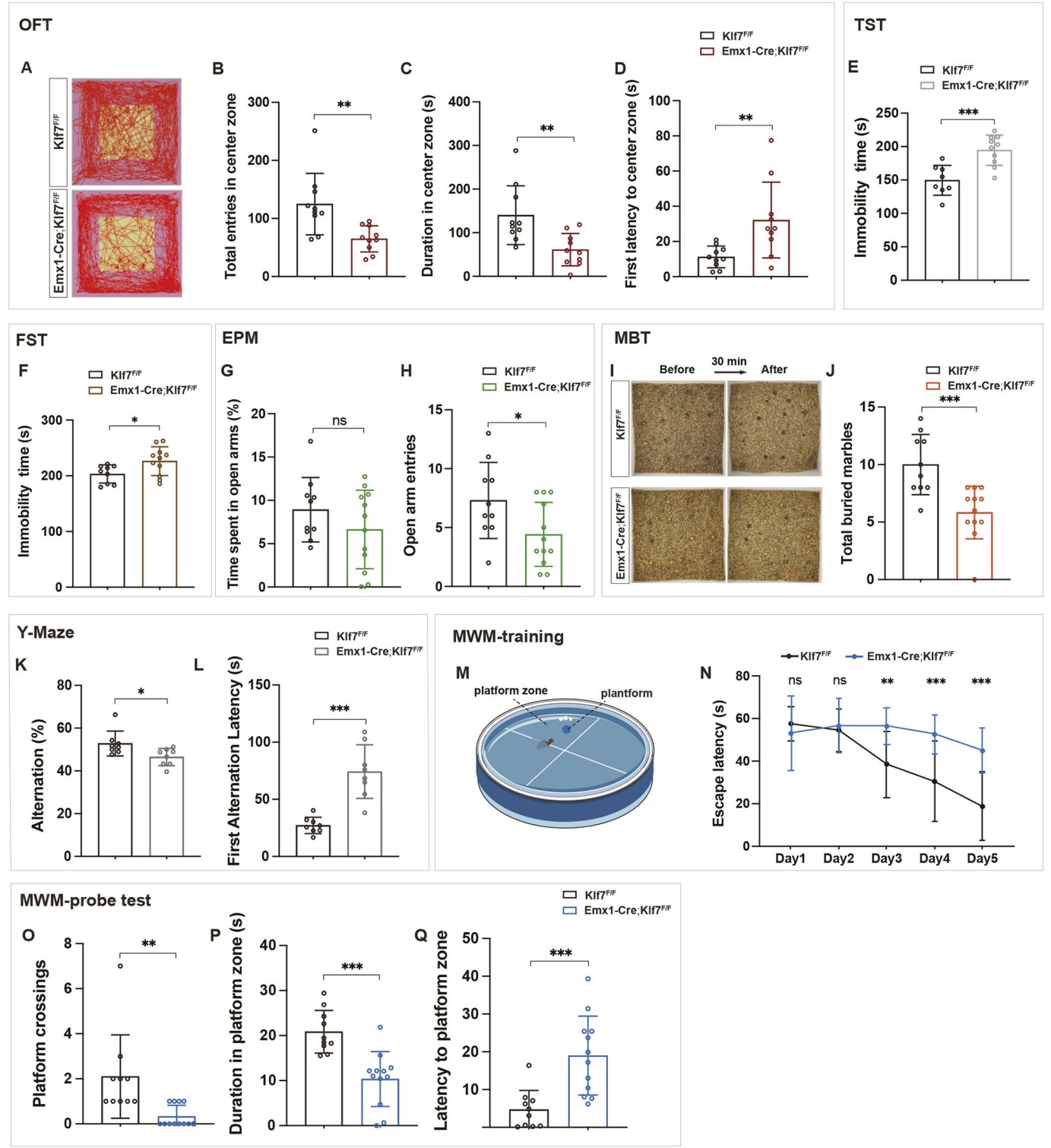

**Fig. 8. KLF7 cKO affects adult behavior.** (A-D) Open field test (OFT) movement analysis (*n*=10). (E,F) Tail suspension test (TST) and forced swim tests (FST) (*n*=8-11). (G,H) Elevated plus maze (EPM) test (*n*=10-12). (I,J) Marble burying test (MBT) (*n*=10-12). (K,L) Y-maze test (*n*=8). (M-Q) Morris Water Maze (MWM) learning and memory test (*n*=10-12). ns, not significant; *$P<0.05$; **$P<0.01$; ***$P<0.001$ [unpaired two-tailed *t*-test (A-M,O-Q), two-way ANOVA followed by Sidak's multiple comparisons test (N)]. Data are mean±s.e.m.

importance in forebrain development. Our recent work established that KLF7 is required for corpus callosum formation and cortical neuron migration by regulating downstream effectors, including p21 and Rac3 (Hong et al., 2023). Here, we further demonstrate that targeted deletion of KLF7 in neural progenitor cells leads to severe hippocampal atrophy and a significant reduction in neuronal numbers in the CA and DG regions, underscoring its pivotal role in forebrain development and hippocampal formation.

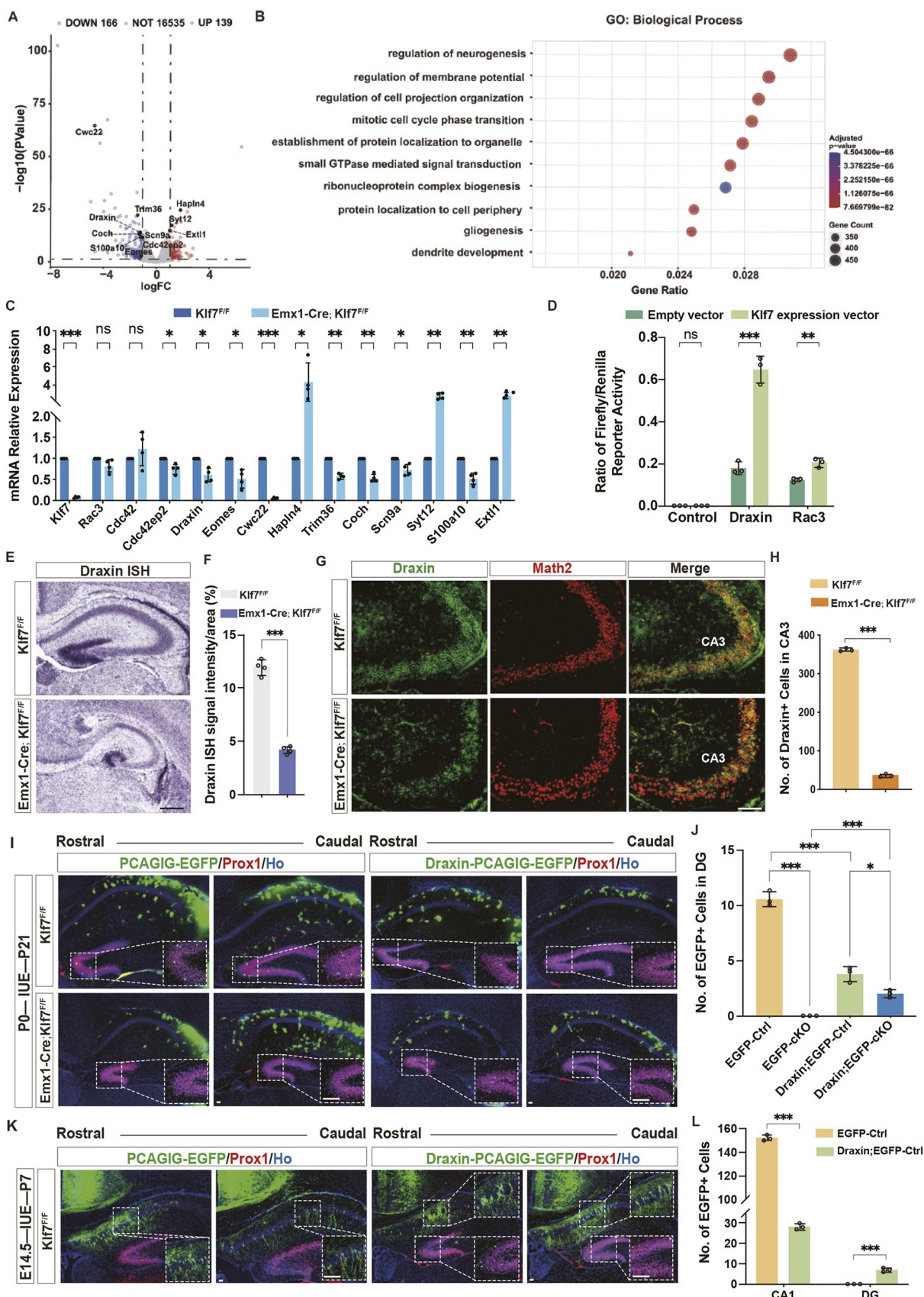

**Fig. 9. KLF7 regulates neuronal migration via Draxin.** (A) Volcano plot of differentially expressed genes (DEGs) in *Emx1-Cre;Klf7*^F/F mice. (B) GO analysis of DEGs. (C,D) RT-qPCR and luciferase assay validating candidate genes. (E-H) Draxin *in situ* hybridization, immunofluorescence and quantification at P7 (*n*=3). (I-L) *In utero* electroporation of Draxin and neuronal migration analysis at P7/P21 (*n*=3). ns, not significant; *P<0.05; **P<0.01; ***P<0.001 [unpaired two-tailed *t*-test (A-I,L), two-way ANOVA followed by Sidak's multiple comparisons test (J)]. Data are mean±s.e.m. Scale bars: 100 μm (E,G); 50 μm (I,K).

Our findings showed that the specific deletion of KLF7 did not affect the distribution and proliferation of progenitor cells in the E14.5 hippocampal primordium. However, by E18.5, significant reductions in precursor cells, including NPCs and IPCs, were observed in the HNE, 1ry and 2ry GM regions. Postnatally, a marked decrease in IPCs was noted in the DG. Further analysis revealed that KLF7 deletion impairs neural progenitor cell differentiation and migration at distinct stages and regions during hippocampal development. The absence of KLF7 disrupts the exit of neural progenitor cells from the cell cycle and their subsequent differentiation, consistent with the established expression pattern of KLF7, which is predominantly observed post-mitosis (Laub et al., 2001). Supporting this, Caiazzo et al. demonstrated via short hairpin RNA-mediated gene silencing that KLF7 expression is essential for the differentiation of neuroectodermal and mesodermal cells *in vitro* (Caiazzo et al., 2010).

During hippocampal development, NPCs migrate from the HNE to the CA pyramidal layer, with peak migration occurring between E14.5 and E16.5 (Fan et al., 2018; Nakahira and Yuasa, 2005). Similarly, DG granule neurons undergo two waves of migration: first, NPCs migrate from the 1ry GM to form the 2ry GM (the DMS) by E15.5, and later into the developing DG granule cell layer, visible by E17.5-E18.5. The second wave involves the migration of NPCs to the 3ry GM, forming ∼85% of DG granule cells during late embryonic and early postnatal stages, with peak proliferation occurring at P1 (Radic et al., 2017). In this context, our results indicate that KLF7 deletion disrupts NPC migration during crucial periods, including their movement from the HNE to the CA pyramidal layer and from the DNE to the DG granule cell layer. These findings suggest that KLF7 plays sequential and multifaceted roles in hippocampal development, guiding cells from progenitor stages to mature neuronal differentiation.

Previous research has shown that KLF7 deficiency impairs cortical efferent projections entering the internal capsule (Laub et al., 2005). Similarly, we observed abnormal accumulation of EdU-labeled migratory neurons in the retrosplenial granular cortex of KLF7 mutant mice and an inability of EGFP-labeled fibers to cross the midline. These findings suggest potential defects in anterior commissure and corpus callosum formation, warranting further investigation. Additionally, we noted structural abnormalities in CA1 PNs and DG granule cells in *Emx1-Cre;Klf7*$^{F/F}$ mice, including reduced dendritic complexity and spine density. While previous studies implicated KLF7 in dendritic arborization of CA1 neurons in KLF7-null mice (Laub et al., 2005), our findings highlight its role in shaping hippocampal dendritic development in a cell-specific context.

The expansion of dendritic arbors is essential for the maturation of neuronal circuits (Forrest et al., 2018). Within the hippocampus, 79% of dCA1 synaptic inputs originate from intrahippocampal projections, primarily from ipsilateral and contralateral CA regions (Tao et al., 2021). CA3 provides dense inputs to CA1 through the Schaffer collateral and commissural pathways, supplemented by inputs from the ENT (Alexandre et al., 2006; Kohara et al., 2014; Ropireddy et al., 2012). dCA1 projection neurons preferentially receive inputs from rostral contralateral CA1, whereas caudal CA1 inputs are sparse, forming essential links in the trisynaptic circuit (Tao et al., 2021). Consistent with previous findings, we observed that dCA1 projection neurons in *Klf7*$^{F/F}$ control mice primarily received inputs from ipsilateral and contralateral CA regions. In contrast, Emx1-Cre; *Klf7*$^{F/F}$ mutant mice displayed disrupted inputs, with dCA1 neurons receiving abnormal projections from the DG and lacking contralateral hippocampal inputs. Furthermore,

projections from cortical regions, including ECT, PRh and ENT, to dCA1 were also disrupted in mutants. Whether these aberrant projections result from dendritic spine defects or forebrain structural anomalies remains unclear and merits further investigation.

The hippocampus is widely recognized as essential for encoding spatial working and reference memory (Lisman et al., 2017; Morellini, 2013). Our behavioral analyses revealed that *Emx1-Cre; Klf7*$^{F/F}$ mutant mice exhibit depressive- and anxiety-like behaviors, as well as memory impairments, underscoring the role of Klf7 in hippocampal function. Previous studies linked Klf7 deficiency to rhythmic gene disruption and autistic behaviors (Tian et al., 2022a). Our findings expand on this by demonstrating the crucial role of KLF7 in hippocampal development and its broader implications for spatial memory, locomotion, arousal and emotional regulation (Engin and Treit, 2007; Loganathan et al., 2024; Pronier et al., 2023; Robinson et al., 2019).

To generate cKO mice, male *Emx1-Cre;Klf7*$^{F/F}$ mice were mated with female *Klf7*$^{F/F}$ mice. This strategy efficiently yields experimental mutant (*Emx1-Cre;Klf7*$^{F/F}$) and control (*Klf7*$^{F/F}$) littermates. We acknowledge that using mutant males as breeders could potentially introduce subtle effects on offspring development or behavior. To rigorously control for this possibility and all background variables, all experimental comparisons, including behavioral analyses, were conducted exclusively between littermates. While no overt phenotypic differences were observed in breeders, the potential contribution of this strategy to the observed phenotypes, particularly behavior, cannot be entirely excluded.

One of the most significant findings of our study is that KLF7 controls the migration of granule cells in the DG by regulating the expression of *Draxin*. To the best of our knowledge, this study is the first to establish a direct relationship between the transcription factor KLF7 and the *Draxin* gene. Draxin, a neural chemorepellent first identified via signal sequence trap, is essential for proper axonal navigation, as evidenced by severe neural circuit defects observed in Draxin-knockout mice (Ahmed et al., 2011; Islam et al., 2009; Shinmyo et al., 2015). It also acts in concert with Tsukushi to guide forebrain commissure formation (Hossain et al., 2013) and interacts with netrin 1 and its receptors DCC and neogenin (Ahmed et al., 2011; Shinmyo et al., 2015), suggesting roles in both direct and netrin 1-modulated guidance pathways.

Beyond axon guidance, Draxin plays a crucial role in hippocampal development. It has been shown to regulate postnatal neurogenesis and granule cell differentiation in the DG (Tawarayama et al., 2020), and its deletion leads to reduced DG volume and fewer granule cells (Tawarayama et al., 2018). In our study, overexpression of Draxin specifically influenced neuronal migration in the DG, with little effect on the CA region, underscoring its regional specificity. Furthermore, Draxin has been implicated in modulating canonical Wnt signaling (Hutchins and Bronner, 2018; Miyake et al., 2012). In zebrafish, a homolog of Draxin acts as a Wnt antagonist by interfering with Wnt-Lrp6 interactions (Miyake et al., 2012), suggesting that reduced Draxin expression could disrupt Wnt-mediated control of cell differentiation and migration. However, further studies are needed to confirm this mechanism in mammals.

Importantly, we demonstrated that Draxin expression was markedly reduced in KLF7 mutant mice, whereas its promoter activity was significantly enhanced by KLF7 overexpression. These findings suggest that KLF7 regulates Draxin to control the migration of granule cells in the DG. Taken together, our results provide compelling evidence that KLF7 plays an essential role

in hippocampal neurogenesis and function, expanding our understanding of its regulatory mechanisms and contributions to hippocampal development.

## MATERIALS AND METHODS
### Mice
All animals were housed in a controlled animal facility with a 12 h light/12 h dark cycle and a room temperature maintained at 23±2°C. Both male and female mice had *ad libitum* access to food and water. The following transgenic mouse lines were used: *Emx1-Cre* (B6.129S2-Emx1tm1-cre-Krj/J, The Jackson Laboratory, J005628) and *Klf7*F/F (generated from Cyagen company). *Emx1-Cre* and *Klf7*F/F were maintained on a C57BL/6 background, as previously described (Gorski et al., 2002; Hong et al., 2023). These strains were intercrossed over multiple generations to generate *Emx1-Cre; Klf7*F/F mice. For all experiments, both male and female embryos and neonatal mice were randomly assigned. The day of plug detection was designated as E0.5, and the day of birth was defined as P0.5. All experimental protocols were approved by the Animal Care and Use Committee of the Shanghai Medical College of Fudan University.

### Breeding assay and genotyping of mice
Twelve- to sixteen-week-old mice were selected for breeding assays with a male-to-female ratio of 1:2. *Emx1-Cre;Klf7*F/F males were crossed with *Klf7*F/F females to generate experimental animals; no breeding pairs consisted of two *Emx1-Cre;Klf7*F/F mice. Successful mating was confirmed by the presence of cervical mucus plugs in all females. Genomic DNA (gDNA) was extracted from tail or digit biopsies using the MightyAmp™ Genotyping Kit (Takara) according to the manufacturer's instructions. The extracted DNA was then used as a template for PCR reactions with the primer pairs listed in Table S1.

### In situ hybridization
*In situ* hybridization was performed as previously described (Qin et al., 2011). Briefly, a 753-bp fragment of mouse cDNA was isolated by PCR from the KLF7 plasmid using the primers listed in Table S1 and subcloned into a pGEM-Teasy expression vector (Promega). An 892-bp fragment of mouse cDNA was PCR-amplified from the Draxin plasmid using the primers listed in Table S1 and subcloned into a pGEM-Teasy expression vector (Promega). Digoxigenin-labeled sense and antisense riboprobes were synthesized through *in vitro* transcription using SP6 and T7 RNA polymerases (Roche), respectively. The *in situ* hybridization procedure was carried out on 40 µm thick coronal cryostat sections of brain tissue.

### RT-PCR and real-time qPCR assays
Total hippocampal RNA from mice at 1 day and 7 days of age was extracted using TRIzol® Reagent (Invitrogen) and reverse-transcribed into cDNA using the PrimeScript™ RT Reagent Kit (Takara) according to the manufacturer's protocol. Real-time PCR was then performed to quantify the expression of specific hippocampal markers using TB Green® Premix Ex Taq™ (Takara) and the primers listed in Table S1, with hippocampal cDNAs as templates.

### Immunohistochemistry
Nissl staining was performed using Cresyl Violet following established protocols (Beyotime). Immunohistochemistry was conducted as previously described (Liu et al., 2023). Briefly, brain sections were washed three times with phosphate-buffered saline (PBS) for 10 min each. The sections were then permeabilized with 0.5% Triton X-100 for 20 min and blocked with 5% bovine serum albumin (BSA) in PBS at room temperature for 1 h. Primary antibodies were incubated overnight at 4°C. Immunohistochemical staining was performed using the following primary antibodies: Ctip2 (rabbit, 1:1000, Abcam, ab240636), Math2/NeuroD6 (rat, 1:1000, Abcam, ab85824), NeuroD (mouse, 1:500, Abcam, ab205300), PAX6 (guinea pig, 1:500, Oasis, OB-PGP078-01), TBR2 (rabbit, 1:500, Abcam, ab183991), PCNA (mouse, 1:500, Santa Cruz Biotechnology, SC-25280), Vglut1 (guinea pig, 1:500, Millipore, ab5905), Synaptoporin (rabbit, 1:500, SYSY, 102002-SYSY), Draxin (sheep, 1:100, R&D Systems, AF6148-SP), Prox1

(mouse, 1:500, Proteintech, 67438-1-lg). After washing, sections were incubated with the corresponding secondary antibodies, conjugated with Alexa Fluor 594, 488 or 647 (1:500, Jackson ImmunoResearch, 715-585-150, 715-545-150, 715-607-003), for 2 h at room temperature. Nuclei were counterstained with Hoechst 33342 (1 µg/ml, Sigma-Aldrich) for 15 min at room temperature. Photomicrographs were captured using a laser scanning confocal microscope (Confocal Leica SP8 inverted) and an Invitrogen EVOS imaging system (EVOS M7000).

### Quantification analysis of hippocampal volume
To quantify the hippocampal volume, brain sections were stained with Nissl and imaged using the EVOS M7000 system. The length and area of the hippocampus, including both the DG and CA regions, were measured along their internal boundaries.

### EdU labeling
Pregnant mice at different embryonic stages were injected with 5-ethynyl-2′-deoxyuridine (EdU, Sigma-Aldrich) at a dose of 50mg/kg body weight. EdU was prepared at a concentration of 1 mg/ml in PBS for injection. Offspring were collected at various time points in accordance with the experimental requirements.

### In utero electroporation and P0.5 electroporation
IUE was performed as previously described (Qin and Zhang, 2012). Briefly, pregnant mice at E14.5 were anesthetized with pentobarbital sodium and xylazine (Sigma-Aldrich), and their uterine horns were exposed. A solution containing plasmid DNA (2.5 µg/µl or 1.5 µg/µl) and Fast Green dye (Sigma-Aldrich) was injected into the lateral ventricles of the embryos using a glass micropipette, with a total volume of 2 µl per embryo. Electroporation was conducted by applying five 50-ms pulses at 34 V with 950-ms intervals using an ECM 830 electroporator (BTX Harvard Apparatus). Following electroporation, the uterine horns were carefully replaced into the abdominal cavity, and the surgical incision was sutured. The brains of the embryos were analyzed at P7. For P0.5 electroporation, a solution containing plasmid DNA (2.5 µg/µl or 1.5 µg/µl) and Fast Green dye (Sigma-Aldrich) was injected into the lateral ventricles of the offspring using a glass micropipette, with a total volume of 2 µl per embryo. Electroporation was conducted by applying five 50-ms pulses at 90 V with 950-ms intervals using an ECM 830 electroporator. Following electroporation, the pups were carefully placed on a temperature-controlled heating pad at 37°C for optimal recovery before being returned to their home cage. The brains of were analyzed at P21.

### Golgi-cox staining and dendritic spine count
The morphology of neuronal dendrites and dendritic spines in the mouse brain was examined using Golgi-Cox staining, following the protocol provided with the FD Rapid GolgiStain Kit (FD NeuroTechnologies). Fresh brain tissue was processed according to the manufacturer's instructions. Dendrites within the CA and DG subregions of the hippocampus were imaged using the EVOS M7000 optical microscope. Dendritic spines along secondary dendrites of the CA1 region were quantified, starting from their origin on primary dendrites. Counting was performed by an experimenter unaware of the group allocation of each sample.

### Virus preparation and stereotaxic brain injection
We used the rabies virus (RV)-based monosynaptic tracing system to identify the monosynaptic inputs of PNs in the dCA1 region. The viruses used in this study were obtained from Lifespan Company (Shanghai, China). An adeno-associated viral vector (AAV) serotype 2-retro (AAV2-Retro) was employed for retrograde labeling and gene expression. The AAV2-Retro vector, carrying the hSyn promoter driving the expression of EGFP fused with P2A and 3XFLAG tags (pAAV-hSyn-EGFP-P2A-3XFLAG), was selected for its ability to efficiently transduce neurons through retrograde transport from axon terminals to cell bodies. All viruses were stored at −80°C until use.

For trans-synaptic retrograde tracing, mice were anesthetized with 2.0% isoflurane for 1 h and secured on a stereotaxic apparatus (RWD, Shenzhen, China). Ointment was applied to their eyes to prevent dryness. Following

iodine disinfection, a midsagittal incision was made to expose the parietal skull. A bone window was created at the designated site, and viral injections were performed using a pulled glass micropipette with a volume of 300 nl. The injection rate was set to 1 nl/s to minimize tissue damage. Following each injection, the micropipette was left in place for 10 min to ensure proper diffusion before withdrawal. To minimize post-operative discomfort, lincomycin-lidocaine gel was applied to the brain surface before suturing. Mice were placed on electric heating pads until fully recovered from anesthesia. After recovery, the animals were returned to their home cages and they remained there until euthanasia 2 weeks later. The injection coordinates, based on the Allen Mouse Brain Atlas, were as follows: dCA1 in Control mice (AP: −1.65 mm; ML: −1.20 mm; DV: −1.52 mm), and dCA1 in KLF7 mutant mice (AP: −1.69 mm; ML: −1.35 mm; DV: −1.52 mm); $n$=3 mice per group.

### Behavioral paradigms
All behavioral assays were conducted when the offspring were 6 months old. Tests were recorded using EthoVision XT 13 software between 09:00 and 16:30, and data were analyzed in an anonymous manner. Mice were habituated to the testing room for 24 h before each experiment. The apparatuses were cleaned with 70% ethanol followed by water between trials to prevent olfactory cues from influencing behavior.

### Open field test and locomotor activity measurement
Mice were placed in open field chambers (40 cm×40 cm) equipped with light beams and allowed to explore freely for 30 min. Sessions were digitally recorded and analyzed using the automated Open Field software. The arena was divided into two zones: the center zone (20 cm×20 cm) and the peripheral zone. Parameters such as total distance traveled, frequency of entries into the center zone, distance covered within the center zone, and time spent in the center zone were measured to assess both locomotor activity and anxiety levels.

### Tail suspension test and forced swimming test
The TST and FST were performed to assess depressive-like behaviors in the offspring, following established protocols. In the TST, individual mice were suspended by their tails on an apparatus for 5 min. In the FST, mice were placed in a transparent cylinder filled with water to a height of 18 cm, at a temperature of 18-26°C, and recorded for 5 min. Behavioral responses during the 5-min sessions of both tests were analyzed.

### Elevated plus maze test
The EPM test was conducted to assess anxiety-like behaviors in the offspring. Mice were placed in the central area of the EPM, which consists of arms measuring 35×5 cm, with closed arms enclosed by 20 cm-high black walls. Their movements were recorded for 5 min using a video camera. The tracking software automatically quantified parameters such as total distance traveled, number of entries, and time spent in the open arms.

### Marble burying test
The MBT was used to assess anxiety- and compulsive-like behaviors. A transparent Plexiglas box (35 cm long×35 cm wide×50 cm high) was filled with 5 cm of standard wood shavings. Sixteen glass marbles were arranged in a 4×4 grid and evenly distributed on the surface of the shavings. Mice were individually placed in the same corner of the box for a 30-min test session. At the end of the session, an image of the marbles was captured. The number of buried marbles was counted from digital images and videos by an investigator unaware of the group allocations.

### Y-maze test
The Y-maze test was used to assess short-term spatial learning and memory. The apparatus consisted of three black arms (A, B and C) arranged at 120° angles to each other. Each arm measured 45 cm in length, 8 cm in width and 30 cm in height. Mice were habituated to the apparatus for 30 min before testing. During the test, each mouse was placed at the center of the Y-maze and allowed to explore freely for 10 min. The number of alternations and arm entries were recorded. An alternation was defined as a sequence of consecutive entries into different arms (e.g. ABC or CBA, but not ABA).

The percentage of alternations was calculated using the formula: [number of alternations/(total number of arm entries−2)]×100%. After each trial, the maze was cleaned with 75% alcohol to prevent olfactory cues from influencing subsequent trials.

### Morris Water Maze test
Spatial learning and memory were assessed using the Morris Water Maze task. The apparatus consisted of a circular pool (1.2 m in diameter, 0.6 m in height) filled with water to a depth of 0.4 m, maintained at a temperature of 22-26°C. In the navigation test, mice were introduced sequentially into the water from four different starting positions, and the latency to escape onto the submerged platform was recorded (with a maximum swimming time of 60 s). After each trial, mice remained on the platform for 10 s. In the probe trial, the quadrant containing the hidden platform was designated as the 'target quadrant' and a single 60 s probe trial was conducted. The number of crossings over the platform's former location and the percentage of time spent in the target quadrant were recorded. All procedures were monitored using a video tracking system.

### RNA-sequencing
Total RNA was extracted from individual hippocampi using TRIzol reagent (Invitrogen). Library preparation and sequencing were performed by Wuhan Frasergen Company. Raw sequencing reads were initially processed to remove adaptors and low-quality bases. Clean reads were then aligned to the mouse genome (GRCm38.p6, ENSEMBL) using HISAT2. For gene expression quantification, read counts were normalized to FPKM values. Fold changes (FCs) were calculated for all pairwise comparisons, and differential gene expression was identified using EBSeq with the criteria of |log2(FC)|>1 and FDR<0.05. Functional annotation of DEGs (FDR<0.05) was performed using the R package for GO enrichment analysis, along with the online bioinformatics tool DAVID (version 6.8).

### Plasmid construction
For IUE and P0.5 electroporation, we constructed the Draxin overexpression plasmid using the pCAG-IRES-EGFP vector as the backbone. The full-length coding sequence (CDS) of the *Draxin* gene was amplified from cDNA derived from mouse brain tissue using primers that included ClaI and XhoI restriction enzyme recognition sites at the 5′ and 3′ ends, respectively (primer sequences are listed in Table S1). The PCR product was gel-purified, and the linearized pCAG-IRES-EGFP vector was digested with ClaI and XhoI. The *Draxin* CDS and the vector were ligated using seamless ligase, and the ligation mixture was transformed into competent *Escherichia coli* cells. Positive clones were selected on ampicillin-containing agar plates. Plasmid DNA was extracted from selected colonies and verified by restriction enzyme digestion and Sanger sequencing to confirm correct insertion and orientation of the *Draxin* CDS.

For luciferase reporter assays, the promoter regions of the mouse *Draxin* (2280 bp) and *Rac3* (2179 bp) genes were cloned separately into the pGL4.23 vector (Promega). Promoter sequences of mouse *Draxin* and *Rac3* were amplified by PCR and inserted into the pGL4.23 vector at the EcoRV and XhoI sites for *Draxin*, and at the ClaI and XhoI sites for *Rac3*, resulting in the constructs pGL4.23-Draxin and pGL4.23-Rac3, respectively. Primer sequences are listed in Table S1. All constructs were confirmed by sequencing.

### Cell culture and transfection
HEK293T cells were maintained in Dulbecco's Modified Eagle Medium (DMEM) with high glucose (Thermo Fisher Scientific), supplemented with 10% fetal bovine serum (FBS, Thermo Fisher Scientific). The cells were cultured at 37°C in a humidified incubator with 5% $CO_2$. Transfection was carried out using the PolyJet *In Vitro* DNA Transfection Reagent (SignaGen Laboratories) according to the manufacturer's instructions.

### Luciferase reporter assay
HEK293T cells were gently lysed 48 h post-transfection and luciferase activities were measured using the Dual-Luciferase Reporter Assay System (Promega). Both firefly and *Renilla* luciferase activities were quantified, with *Renilla* luciferase serving as an internal control for normalizing transfection efficiency across samples. Each assay was performed in

triplicate to ensure reproducibility, and the entire experiment was independently repeated at least twice to validate the results.

## Quantification and statistical analysis

ImageJ was used for image analysis, while GraphPad Prism 9 was employed for statistical data analysis. Data from independent groups were presented as mean±s.e.m. based on at least three independent experiments. Statistical comparisons were performed using unpaired two-tailed Student's $t$-tests or one-way ANOVA, as appropriate. Data from Fig. 8N and Fig. 9J were analyzed using two-way ANOVA followed by Sidak's multiple comparisons test. Detailed statistical information for each experiment is provided in the figure legends. Throughout the figure legends and results sections, statistical significance is indicated as follows: $*P<0.05$, $**P<0.01$ and $***P<0.001$.

## Acknowledgements
We thank members of the Qin laboratory for suggestions and comments. We thank the Imaging Core Facility of the State Key Laboratory of Medical Neurobiology at Fudan University for the support.

## Competing interests
The authors declare no competing or financial interests.

## Author contributions
Conceptualization: Y.L., S.Q.; Data curation: Y.L., W.H., Y.Z., A.Z., G.Q., Z.W., Xuanming Shi, S.Q.; Formal analysis: Y.L., Y.Z., A.Z., C.Q., S.Q.; Funding acquisition: S.Q.; Investigation: Y.L., W.H., P.G., G.Q., Xuan Song, S.Q.; Methodology: Y.L., W.H., Y.Z., P.G., Xuan Song, Z.W., Xuanming Shi, C.Q., S.Q.; Project administration: S.Q.; Resources: S.Q.

## Funding
This work was supported by the National Natural Science Foundation of China (31871477 and 32170971), the Natural Science Foundation of Shanghai Municipality (18ZR1403800) and the National Key Research and Development Program of China (973 Program, 2014CB965001). Open access funding provided by Fudan University. Deposited in PMC for immediate release.

## Data and resource availability
All relevant data can be found within the article and its supplementary information.

## Peer review history
The peer review history is available online at https://journals.biologists.com/dev/lookup/doi/10.1242/dev.204718.reviewer-comments.pdf

## Special Issue
This article is part of the Special Issue 'Lifelong Development: the Maintenance, Regeneration and Plasticity of Tissues', edited by Meritxell Huch and Mansi Srivastava. See related articles at https://journals.biologists.com/dev/issue/152/20.

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
