## [Peer Review File · Development (Cambridge, England)]

KLF7 orchestrates hippocampal development through neurogenesis and Draxin-mediated neuronal migration

Yitong Liu, Wentong Hong, Yuyan Zhou, Ailing Zhang, Pifang Gong, Guibo Qi, Xuan Song, Zhenru Wang, Xuanming Shi, Congcong Qi and Song Qin
DOI: 10.1242/dev.204718

Editor: Debra L. Silver

Review timeline

Original submission:	20 February 2025
Editorial decision:	21 March 2025
First revision received:	6 May 2025
Editorial decision:	30 May 2025
Second revision received:	23 June 2025
Accepted:	26 June 2025

Original submission

First decision letter

MS ID#: dev.204718

MS TITLE: KLF7 orchestrates hippocampal development through neurogenesis and Draxin-mediated neuronal migration

AUTHORS: Yitong Liu, Wentong Hong, Yuyan Zhou, Ailing Zhang, Pifang Gong, Guibo Qi, Xuan Song, Zhenru Wang, Xuanming Shi, Congcong Qi and Song Qin

Dear Dr Qin,

I have now received all the referees' reports on the above manuscript, and have reached a decision. The referees' comments are appended below, or you can access them online: please go to:

As you will see, the referees express considerable interest in your work, but have some significant criticisms and recommend a substantial revision of your manuscript before we can consider publication. Please address concerns about supporting references, interpretations of findings including possible discrepancies, and rigor and validation of results (ISH and IF, cell death, etc). Please also be sure that any points of title and abstract are well supported, as suggested by Reviewer 2. If you are able to revise the manuscript along the lines suggested, which will involve further experiments, I will be happy to receive a revised version of the manuscript. Your revised paper will be re-reviewed by one or more of the original referees, and acceptance of your manuscript will depend on your addressing satisfactorily the reviewers' major concerns. Please also note that Development will normally permit only one round of major revision. If it would be helpful, you are welcome to contact us to discuss your revision in greater detail. Please send us a point-by-point response indicating your plans for addressing the referees' comments, and we will look over this and provide further guidance.

Please attend to all of the reviewers' comments and ensure that you clearly highlight all changes made in the revised manuscript. Please avoid using 'Tracked changes' in Word files as these are lost in PDF conversion. I should be grateful if you would also provide a point-by-point response detailing

how you have dealt with the points raised by the reviewers in the 'Response to Reviewers' box. If you do not agree with any of their criticisms or suggestions please explain clearly why this is so.

Reviewer 1

Advance summary and potential significance to field

Liu et al. characterized the phenotypes of Emx1-Cre Klf7 KO mice. The results demonstrate that KLF7 loss leads to abnormal hippocampal development and connectivity. Gene expression profiling and genetic rescue experiments suggest that Draxin plays a role in mediating the aberrant granule neuron migration observed in KO mice. The study is well-executed, with appropriate experimental approaches and well-presented figures supported by quantifications. The study will increase our understanding of the role of KLF7 in hippocampus development.

Comments for the author

However, several major concerns need to be addressed to improve the manuscript's clarity and strengthen its conclusions.

Major Concerns

1. One of the key conclusions is that KLF7 loss disrupts NPC cell cycle progression and differentiation dynamics, as inferred from EdU labeling and PCNA expression. Several points require clarification:

- What is the normal cell cycle length of hippocampal NPCs at these developmental stages?

Supporting literature is needed.

- The claim that "cells labeled with EdU but not PCNA were considered progenitors that had exited the cell cycle" assumes continuous proliferation with a short interphase. Is there literature support for this? If not, the interpretation of Section 3.3 should be reconsidered.

- The data in Section 3.3 suggest increased NPC proliferation and reduced differentiation into IPCs. Additional analyses would clarify and strengthen these findings:

- a) What is the number of PCNA+ cells in the HNE and germinative matrices?

- b) What is the proliferation index of NPCs and IPCs (% PCNA+ PAX6+ and % PCNA+ TBR2+ cells)?

- c) Could altered cell death contribute to the observed phenotype? Cell death analysis is recommended.

2. Figure 5N and O: If few CA neurons and no DG neurons originate from the VZ, how do the CA and DG regions form in the KO?

3) Another major finding is that KLF7 functions through Draxin, but several concerns need to be addressed:

- Draxin ISH (Figure 9E): The images do not convincingly show reduced Draxin expression in the KO. Quantification of the ISH signal in the CA and DG is necessary.

- Draxin IF (Figure 9F): The current figure focuses on CA3. High-magnification images of Draxin localization in CA1 and DG would provide more insight into the DG migration rescue phenotype.

- How does Draxin, as a chemorepellent, regulate granule neuron migration? In which hippocampal regions (CA, DG, or other cell types) does Draxin function? The discussion should be expanded using current literature on granule neuron migration. Additionally, the pleiotropic effects of Draxin overexpression (Figure 9I and K) should be acknowledged.

- The identity of GFP+ cells in the DG (Figure 9H and J) should be confirmed using granule neuron markers such as PROX1.

Minor Concerns

1) The manuscript refers to "Emx1-Cre" as a driver of "anterior neural progenitor cells" and "neural progenitor-specific" recombination. It should be clarified that Emx1-Cre targets progenitor cells whose progeny include forebrain excitatory neurons and glia (astrocytes and oligodendrocytes).

2) When discussing KLF7-related neurodevelopmental conditions in humans, "pathogenic variants" is preferred over "mutations."

3) Methods

- Breeding Assay:

a) The stated male-to-female ratio of 2:1 should be verified, as this is not typically recommended.
b) The breeding strategy should be clarified: Using Emx1-Cre; Klf7F/F (KO) mice as breeders could introduce developmental phenotypes in offspring due to impaired parental care, given the KO's behavioral deficits. Typically, Emx1-Cre; Klf7F/+ heterozygous mice are used as breeders to avoid confounds.

- Plasmid Construction:

In the luciferase reporter assay, why were full-length Draxin and Rac3 cDNAs used? Typically, only the promoter region is cloned into the reporter vector.

- Luciferase Reporter Assay:

Firefly luciferase should not be used as an internal control; Renilla luciferase is the standard for normalizing transfection efficiency.

4) Figures & Data Presentation

- Figure 4: Additional X-axis labels should be added to panels C, D, E, H, I, J, M, N, and O to indicate both the EdU injection and tissue collection timepoints. Currently, only the collection time is labeled (e.g., "24 hours later"), which is confusing when interpreting statements such as "At E15.5, Klf7-deficient mice showed a decreased tendency for progenitor cells to exit the cell cycle." Since EdU was injected at E15.5, but data is displayed for E16.5, this should be clarified in the figure.

- Section 3.4: The statement, "At corresponding coronal levels, we observed an abnormal accumulation of these markers in the retrosplenial granular (RSG) cortex of Klf7 mutant mice, which was absent in the control group (Figures S2A and S2B)," is inaccurate—Figure S2B shows marker accumulation in the control as well. Please clarify.

- Section 3.6: The manuscript states, "we unilaterally injected a pAAV-hSyn-EGFP-P2A-3XFLAG virus into the dCA1 of 6-month-old adult mice." It should be explicitly mentioned that this is a retrograde AAV (only stated in the Methods), as this is crucial for understanding the experimental design.

- Luciferase Reporter Assay (Figure 9D): The statement that "KLF7 overexpression significantly increased the luciferase activity of pGL4-Draxin, suggesting that KLF7 enhances Draxin mRNA stability" is incorrect. A promoter-reporter assay only assesses transcriptional regulation, not mRNA stability.

Reviewer 2

Advance summary and potential significance to field

The manuscript entitled "KLF7 orchestrates hippocampal development through neurogenesis and Draxin-mediated neuronal migration" investigates the role of Krüppel-like factor 7 (KLF7), an autism spectrum disorder-associated gene, in hippocampus development using a conditional knockout mouse model (Emx1-Cre; Klf7F/F). Unlike Klf7-deficient mice used in previous studies, the conditional knockout mice survived into adulthood and could be analyzed from embryonic development to later adult timepoints. This study brings together analyses at the molecular, cellular, circuit, and behavioral levels, representing a large body of work describing a wide range of phenotypes to better understand hippocampus development, which is unique in many ways and is relatively understudied compared to neocortical development. The manuscript could benefit from a clear, unifying theme that links together the variety of phenotypes. In its current form, the developmental mechanisms underlying the postnatal phenotypes are not clear and the manuscript lacks cohesion.

Comments for the author

1. The title and conclusions suggest that the deficits in hippocampus development after Klf7 knockout are due to deficits in neurogenesis and neuronal migration. Though this could be the case for the dentate gyrus region, the evidence supporting these two mechanisms is not irrefutable for the CA regions. Though dentate gyrus neurogenesis is quite protracted and largely occurs postnatally, neurogenesis in the CA regions of the hippocampus occur embryonically, peaking around E15.5. Figure 1 shows that significant changes in CA1 and CA3 neuron numbers do not occur until postnatal day 7 (P7). I would interpret this as a degeneration phenotype, rather than a neurogenesis phenotype, because of the lag in timing from peak neurogenesis. Was cell death or neuron degeneration ever analyzed or observed in the CA1 and CA3 regions?

2. Figure 2 and 4 suggest that there is an embryonic deficit in neuronal differentiation in the hippocampal neuroepithelium (HNE), the germinal zone for CA1 and CA3 neurons. Figure 4 specifically shows that this deficit occurs around E16.5, but not before or after. It seems strange that a deficit in neuronal differentiation at E16.5 does not lead to changes in neuronal numbers until P7 (Figure 1). I would expect to see reduced neuronal numbers by P0 if there were differentiation defects at E16.5. What is the explanation for this discrepancy?

3. Figure 5 is interpreted as Klf7 knockout causes reduced neurogenesis, but again cell death was not analyzed and remains a possible interpretation for the lower numbers of EdU+ cells or GFP+ cells (in utero electroporation) in the hippocampus at P7.

4. In Figure 5, it is not clear whether the EdU or in utero electroporation initially labels the same number of cells in Control and Klf7 knockout hippocampus. An acute time point would be necessary to determine whether the starting population is the same. This issue will also impact the interpretation of the results, especially IUE at P0, which occurs when there are already phenotypic differences observed in the hippocampus.

5. Figure 6 and 7 show there are clearly defects in the connectivity of the hippocampal neurons in the Klf7 knockout. One of the most striking findings is the complete lack of CA inputs from the contralateral side in the Klf7 knockout. When I saw this result, I thought that maybe the contralateral input is necessary for proper synaptic/circuit integration and might be required for cell survival. Again, I wonder if the neurons are made more or less normally, but do not receive proper input and start to die or shrink postnatally.

6. Why was the P7 timepoint chosen for the RNA-seq experiment in Figure 9? By P7 the hippocampus phenotype of fewer neurons is quite significant. It seems like a better experiment would be to look a bit earlier (P0?) to see what is changing before the neuronal loss occurs.

7. The results in Figure 9 after overexpression of Draxin are confusing. It appears that overexpressing Draxin in Control reduces the number of GFP+ cells, but overexpression in Klf7 knockouts increases the number of GFP+ cells. This discrepancy in the effects of Draxin on the control vs. Klf7 knockout are not explained. I think this is important to have strong data for this experiment because the title claims "Draxin-mediated neuronal migration".

8. It is not clear what data in the manuscript support the conclusion that Klf7 knockout results in neuronal migration defects. It is suggested that the decrease in differentiated cells in the CA and DG regions of the hippocampus (Figure 5) is due to impaired migration, but there is no evidence that these cells get "stuck" in the hippocampal neuroepithelium. Is there evidence that indeed these cells do not migrate and are stuck?

First revisionAuthor response to reviewers' comments**Point-by-point responses to Reviewers' comments****Comments from the Reviewers:**

Reviewer 1: SUMMARY OF THE ADVANCE MADE IN THIS PAPER AND ITS POTENTIAL SIGNIFICANCE TO THE FIELD

Liu et al. characterized the phenotypes of Emx1-Cre Klf7 KO mice. The results demonstrate that KLF7 loss leads to abnormal hippocampal development and connectivity. Gene expression profiling and genetic rescue experiments suggest that Draxin plays a role in mediating the aberrant granule neuron migration observed in KO mice. The study is well-executed, with appropriate experimental approaches and well-presented figures supported by quantifications. The study will increase our understanding of the role of KLF7 in hippocampus development.

Response: We very much appreciate the reviewer's constructive comments and suggestions. The comments are necessary for improving the quality of our manuscript.

1. One of the key conclusions is that KLF7 loss disrupts NPC cell cycle progression and differentiation dynamics, as inferred from EdU labeling and PCNA expression. Several points require clarification:

Comments: - What is the normal cell cycle length of hippocampal NPCs at these developmental stages? Supporting literature is needed.

Response: Thank you for your valuable comments and suggestions. We have revised the text to address your concerns and have clarified the relevant section. Please see page 12, lines 34-37 of the revised manuscript.

Comments: - The claim that "cells labeled with EdU but not PCNA were considered progenitors that had exited the cell cycle" assumes continuous proliferation with a short interphase. Is there literature support for this? If not, the interpretation of Section 3.3 should be reconsidered.

Response: Thank you for the reviewer's comments. For reference, in Figure 2 of a 2021 *Science Advances* paper (<https://www.science.org/doi/abs/10.1126/sciadv.abc6093>), the authors describe: "We labeled proliferating cells by a pulse of 5-ethynyl-2'-deoxyuridine (EdU) and collected the embryos 24 hours later. We then used Ki67 staining to detect proliferating progenitors when the cortex was collected. Cells labeled by EdU, but not Ki67, are the progenitors that have exited the cell cycle (Fig. 2, A and B)."

Ki67 is expressed during the late G1, S, G2, and M phases but is rapidly degraded after mitosis, making it unsuitable for labeling cells in early G1 or quiescent phases. In contrast, PCNA is expressed throughout all stages of the cell cycle. Therefore, to more comprehensively assess cell proliferation, we used EdU labeling in combination with PCNA staining to monitor cell cycle dynamics during embryonic neurogenesis in our study.

- The data in Section 3.3 suggest increased NPC proliferation and reduced differentiation into IPCs. Additional analyses would clarify and strengthen these findings:

a) Comments: What is the number of PCNA+ cells in the HNE and germinative matrices?

b) Comments: What is the proliferation index of NPCs and IPCs (% PCNA+ PAX6+ and % PCNA+ TBR2+ cells)?

Response: We greatly appreciate the reviewer's thoughtful comments regarding points (a) and (b). In Section 3.3, we investigated cell cycle dynamics within a 24-hour timeframe across different stages of embryonic neurogenesis. To achieve this, we employed EdU labeling, which involves administering EdU within a defined time window and analyzing the embryos 24 hours later. This method enables the identification of proliferating cells during the specific interval, making it well-suited for tracking dynamic cell proliferation events. In contrast, PCNA staining reflects the proliferation status at a single time point, which does not align with the objectives of our study.

Additionally, PAX6 is expressed in both quiescent and active NPCs, as well as in IPCs (Figure 1). Some cells exist in a transitional state, co-expressing PAX6 and TBR2. Particularly within the 1ry GM, 2ry GM, and 3ry GM of the developing hippocampus, most cells co-express PAX6 and TBR2. Therefore, EdU+PAX6+TBR2+/EdU+ cells represent progenitors in transition over the 24-hour period, while EdU+PAX6-TBR2+/EdU+ cells indicate cells that have fully differentiated from NPCs into IPCs during the same timeframe (Figure 2).

Given these considerations, and to avoid introducing unnecessary complexity or confusion for readers, we decided not to include analyses related to points (a) and (b). We hope the reviewer agrees that this was the appropriate decision.

Figure 1. PAX6 is expressed in quiescent and active NPCs and IPCs.

NOTE: Figure provided for reviewer has been removed. It showed Figure 5 from Turrero García M., Chang Y., Arai Y. and Huttner W. B. (2016). S-phase duration is the main target of cell cycle regulation in neural progenitors of developing ferret neocortex. *J. Comp. Neurol.* 524, 456-470. doi:10.1002/cne.23801

Figure 2. Different progenitor cell types are distinguished by the expression of PAX6 and/or TBR2.

NOTE: Figure provided for reviewer has been removed. It showed Figure 5 from Ochi S., Manabe S., Kikkawa T. and Osumi N. (2022). Thirty years' history since the discovery of Pax6: From central nervous system development to neurodevelopmental disorders. *Int. J. Mol. Sci.* 23, 6115. doi:10.3390/ijms23116115

c) Comments: Could altered cell death contribute to the observed phenotype? Cell death analysis is recommended.

Response: Thank you for the reviewer's valuable comments. We agree with your suggestion. In our earlier studies, we assessed apoptosis in the developing hippocampus at E14.5 using cleaved caspase-3 staining and observed no abnormal cell death in the hippocampus of *Emx1-Cre;Klf7^{F/F}* mice. Following the reviewers' valuable feedback, we extended our investigation to examine cell apoptosis in the hippocampus at additional time points, including E17.5, P0, and P7. We have revised the manuscript in the 3.3 section to include the following "...we first evaluated apoptotic activity during development. Immunostaining for cleaved caspase-3 revealed no significant difference in programmed cell death between *Klf7*-deficient mice and *Klf7^{F/F}* controls at E14.5. Similarly, cleaved caspase-3-positive cells remained unchanged at E17.5, P0, and P7 (Figures S2A and S2B)". Please see the revised manuscript on page 12, lines 29-33.

2. Comments: Figure 5N and O: If few CA neurons and no DG neurons originate from the VZ, how do the CA and DG regions form in the KO?

Response: Thank you for the reviewer's comments. In utero electroporation (IUE) is a valuable method for studying mammalian brain development in vivo, particularly for investigating neuronal migration and enabling spatial and temporal control of gene expression. However, IUE typically transfects only a subset of neurons and does not label the entire neuronal population. As a result, relatively few EGFP+ cells were observed migrating to the CA and DG regions in the *Emx1-Cre;Klf7^{F/F}* group. This observation suggests a pronounced migration defect in the mutant group compared to controls, but it should not be interpreted to mean that only the detected EGFP+ neurons migrated. A substantial number of untransfected neurons were also likely involved in the migration process but were not labeled by EGFP.

3. Another major finding is that KLF7 functions through Draxin, but several concerns need to be addressed:

Comments: - Draxin ISH (Figure 9E): The images do not convincingly show reduced Draxin expression in the KO. Quantification of the ISH signal in the CA and DG is necessary.

Response: Thanks for the reviewer's valuable comments. As suggested by the reviewer, we have performed the ISH signal analysis to confirm Draxin expression in the mouse hippocampus. The new data are now included in Figure 9F of the revised manuscript.

Comments: - Draxin IF (Figure 9F): The current figure focuses on CA3. High-magnification images of Draxin localization in CA1 and DG would provide more insight into the DG migration rescue phenotype.

Response: Thanks for the reviewer's comments. Figure 9E-H demonstrates that the deletion of KLF7 leads to a marked downregulation of Draxin at both mRNA and protein levels. This finding serves as a complementary validation of the RNA sequencing results, distinct from the DG migration rescue

phenotype. We agree that high-magnification images of Draxin localization in CA1 and DG would provide more insights into the Draxin expression. However, we have encountered technical challenges in obtaining reliable immunofluorescence staining for Draxin in these regions. Despite testing multiple antibody samples, we found that the currently available antibodies for Draxin are not yet fully optimized for immunofluorescence applications. While one antibody provided satisfactory staining in the CA3 region, its performance in CA1 and DG was inconsistent and did not yield reliable or interpretable results. Given these limitations, we focused our analysis on the CA3 region, where the staining was robust and reproducible. We hope that future advancements in antibody development will enable more comprehensive investigations of Draxin localization across all hippocampal subregions.

Comments: - How does Draxin, as a chemorepellent, regulate granule neuron migration? In which hippocampal regions (CA, DG, or other cell types) does Draxin function? The discussion should be expanded using current literature on granule neuron migration. Additionally, the pleiotropic effects of Draxin overexpression (Figure 9I and K) should be acknowledged.

Response: Thank you for the reviewer's valuable comments. Currently, research on Draxin remains limited. Our study is the first to demonstrate, through *in vivo* overexpression experiments, that Draxin regulates the migration of granule neurons. However, the precise mechanisms underlying this regulation remain unclear and require further investigation. While the specific sites of Draxin function within the hippocampus are not fully defined, previous studies, along with our findings, suggest its activity is localized to the DG region. Notably, no prior studies have investigated the effects of Draxin overexpression. To reflect these points and recent advances in understanding granule neuron migration, we have revised the Discussion section to include the following:

"*Draxin*, a neural chemorepellent first identified via signal sequence trap, is essential for proper axonal navigation, as evidenced by severe neural circuit defects observed in *Draxin*-knockout mice (Ahmed et al., 2011; Islam et al., 2009; Shinmyo et al., 2015). It also acts in concert with *Tsukushi* to guide forebrain commissure formation (Hossain et al., 2013) and interacts with *netrin-1* and its receptors *DCC* and *Neogenin* (Ahmed et al., 2011; Shinmyo et al., 2015), suggesting roles in both direct and *netrin-1*-modulated guidance pathways."

"Beyond axon guidance, *Draxin* plays a critical role in hippocampal development. It has been shown to regulate postnatal neurogenesis and granule cell differentiation in the DG (Tawarayama et al., 2020), and its deletion leads to reduced DG volume and fewer granule cells (Tawarayama et al., 2018). In our study, overexpression of *Draxin* specifically influenced neuronal migration in the DG, with little effect on the CA region, underscoring its regional specificity. Furthermore, *Draxin* has been implicated in modulating canonical Wnt signaling (Hutchins and Bronner, 2018; Miyake et al., 2012). In zebrafish, a homolog of *Draxin* acts as a Wnt antagonist by interfering with Wnt-Lrp6 interactions (Miyake et al., 2012), suggesting that reduced *Draxin* expression could disrupt Wnt-mediated control of cell differentiation and migration. However, further studies are needed to confirm this mechanism in mammals."

Please see the revised manuscript on page 20, lines 15-32.

Comments: - The identity of GFP+ cells in the DG (Figure 9H and J) should be confirmed using granule neuron markers such as PROX1.

Response: Thank you for the reviewer's valuable comments. As recommended, we performed immunofluorescence staining for PROX1 to verify the identity of EGFP+ cells in DG. The new results have been incorporated into Figures 9I and 9K of the revised manuscript.

Minor Concerns

1) Comments: The manuscript refers to "*Emx1-Cre*" as a driver of "anterior neural progenitor cells" and "neural progenitor-specific" recombination. It should be clarified that *Emx1-Cre* targets progenitor cells whose progeny include forebrain excitatory neurons and glia (astrocytes and oligodendrocytes).

Response: We sincerely thank the reviewer for pointing this out. We have carefully reviewed the manuscript and updated the description of "*Emx1-Cre*" as follows: "The *Emx1-Cre* transgenic mouse enables genetic recombination in most of neurons of the neocortex and hippocampus, as well as in

a limited population of cortical glial cells, making it a valuable model for studying forebrain development and function (Hong et al., 2023).” Please see page 10, lines 32-35.

2) Comments: When discussing KLF7-related neurodevelopmental conditions in humans, "pathogenic variants" is preferred over "mutations."

Response: Thank you for the reviewer’s helpful comment. As suggested, we have revised the term “mutations” to “pathogenic variants” in the manuscript. These changes have been made on page 2, line 17, and page 3, line 23 of the revised version.

3) Methods

- Breeding Assay:

a) Comments: The stated male-to-female ratio of 2:1 should be verified, as this is not typically recommended.

Response: We sincerely thank the reviewer for careful reading. We are sorry for our careless mistakes. We have revised the following sentence as follows: “Twelve- to sixteen-week-old mice were selected for breeding assays with a male-to-female ratio of 1:2” in the revised manuscript on page 4, line 22.

b) Comments: The breeding strategy should be clarified: Using Emx1-Cre; Klf7F/F (KO) mice as breeders could introduce developmental phenotypes in offspring due to impaired parental care, given the KO’s behavioral deficits. Typically, Emx1-Cre; Klf7F/+ heterozygous mice are used as breeders to avoid confounds.

Response: We thank the reviewer for the valuable comments. Although we did not observe any abnormal developmental phenotypes in the control group among the offspring, we plan to use *Emx1-Cre; Klf7^{F/+}* heterozygous mice for breeding in future experiments. This approach will help enhance the rigor and reliability of our study design.

- Plasmid Construction:

Comments: In the luciferase reporter assay, why were full-length Draxin and Rac3 cDNAs used? Typically, only the promoter region is cloned into the reporter vector.

Response: We thank the reviewer for pointing this out and apologize for the oversight. The sentence has been revised to: “Promoter sequences of mouse *Draxin* and *Rac3* were amplified...” in the revised manuscript on page 9, line 33.

- Luciferase Reporter Assay:

Comments: Firefly luciferase should not be used as an internal control; Renilla luciferase is the standard for normalizing transfection efficiency.

Response: We sincerely thank the reviewer for the careful reading and apologize for the oversight. The error has been corrected in the revised manuscript on page 10, line 12.

4) Figures & Data Presentation

Comments: - Figure 4: Additional X-axis labels should be added to panels C, D, E, H, I, J, M, N, and O to indicate both the EdU injection and tissue collection timepoints. Currently, only the collection time is labeled (e.g., “24 hours later”), which is confusing when interpreting statements such as “At E15.5, Klf7-deficient mice showed a decreased tendency for progenitor cells to exit the cell cycle.” Since EdU was injected at E15.5, but data is displayed for E16.5, this should be clarified in the figure.

Response: Thanks for the reviewer’s comments and suggestions. We have revised Figure 4 according to the reviewer’s suggestion.

Comments: - Section 3.4: The statement, “At corresponding coronal levels, we observed an abnormal accumulation of these markers in the retrosplenial granular (RSG) cortex of Klf7 mutant

mice, which was absent in the control group (Figures S2A and S2B)," is inaccurate—Figure S2B shows marker accumulation in the control as well. Please clarify.

Response: We thank the reviewer for pointing this out. We have revised the sentence to read: "At corresponding coronal levels, *Klf7* mutant mice showed excessive accumulation of these markers in the retrosplenial granular cortex compared to controls (Figures S3A and S3B)," as reflected in the revised manuscript on page 14, lines 32-34.

Comments: - Section 3.6: The manuscript states, "we unilaterally injected a pAAV-hSyn-EGFP-P2A-3XFLAG virus into the dCA1 of 6-month-old adult mice." It should be explicitly mentioned that this is a retrograde AAV (only stated in the Methods), as this is crucial for understanding the experimental design.

Response: We thank the reviewer for pointing this out and fully agree with the suggestion. We have revised the sentence as follows: "We unilaterally injected a pAAV-hSyn-EGFP-P2A-3XFLAG retrograde trans-monosynaptic virus into the dCA1 of 6-month-old adult mice and systematically analyzed the direct inputs to dCA1 PNs," in the revised manuscript on page 15, line 16.

Comments: - Luciferase Reporter Assay (Figure 9D): The statement that "KLF7 overexpression significantly increased the luciferase activity of pGL4-Draxin, suggesting that KLF7 enhances Draxin mRNA stability" is incorrect. A promoter-reporter assay only assesses transcriptional regulation, not mRNA stability.

Response: We thank the reviewer for the helpful reminder. We have corrected the description to read: "KLF7 overexpression significantly increased the luciferase activity of pGL4-Draxin, suggesting that KLF7 positively regulates Draxin promoter activity," as shown in the revised manuscript on page 17, lines 14-15.

Reviewer 2: SUMMARY OF THE ADVANCE MADE IN THIS PAPER AND ITS POTENTIAL SIGNIFICANCE TO THE FIELD

The manuscript entitled "KLF7 orchestrates hippocampal development through neurogenesis and Draxin-mediated neuronal migration" investigates the role of Krüppel-like factor 7 (KLF7), an autism spectrum disorder-associated gene, in hippocampus development using a conditional knockout mouse model (*Emx1-Cre; Klf7^{F/F}*). Unlike *Klf7*-deficient mice used in previous studies, the conditional knockout mice survived into adulthood and could be analyzed from embryonic development to later adult timepoints. This study brings together analyses at the molecular, cellular, circuit, and behavioral levels, representing a large body of work describing a wide range of phenotypes to better understand hippocampus development, which is unique in many ways and is relatively understudied compared to neocortical development. The manuscript could benefit from a clear, unifying theme that links together the variety of phenotypes. In its current form, the developmental mechanisms underlying the postnatal phenotypes are not clear and the manuscript lacks cohesion.

Response: We very much appreciate the enthusiasm and constructive comments of this reviewer. The comments are necessary for improving the quality of our manuscript.

SUGGESTIONS TO AUTHORS

1. Comments: The title and conclusions suggest that the deficits in hippocampus development after *Klf7* knockout are due to deficits in neurogenesis and neuronal migration. Though this could be the case for the dentate gyrus region, the evidence supporting these two mechanisms is not irrefutable for the CA regions. Though dentate gyrus neurogenesis is quite protracted and largely occurs postnatally, neurogenesis in the CA regions of the hippocampus occur embryonically, peaking around E15.5. Figure 1 shows that significant changes in CA1 and CA3 neuron numbers do not occur until postnatal day 7 (P7). I would interpret this as a degeneration phenotype, rather than a neurogenesis phenotype, because of the lag in timing from peak neurogenesis. Was cell death or neuron degeneration ever analyzed or observed in the CA1 and CA3 regions?

Response: Thanks for the reviewer's valuable comments. As we mentioned before, during our earlier studies, we observed cell apoptosis in the developing hippocampus at E14.5 and E16.5 using caspase-3 staining. The findings revealed that no abnormal cell apoptosis was detected in the

hippocampus of *Emx1-Cre;Klf7^{F/F}* mice. Following the reviewers' valuable feedback, we extended our investigation to examine cell apoptosis in the hippocampus at additional time points, including E17.5, P0, and P7. We have revised the manuscript in the 3.3 section to include the following " ...,we first evaluated apoptotic activity during development. Immunostaining for cleaved caspase-3 revealed no significant difference in programmed cell death between *Klf7*-deficient mice and *Klf7^{F/F}* controls at E14.5. Similarly, cleaved caspase-3-positive cells remained unchanged at E17.5, P0, and P7 (Figures S2A and S2B)". Please see the revised manuscript on page 12, lines 29-34.

2. Comments: Figure 2 and 4 suggest that there is an embryonic deficit in neuronal differentiation in the hippocampal neuroepithelium (HNE), the germinal zone for CA1 and CA3 neurons. Figure 4 specifically shows that this deficit occurs around E16.5, but not before or after. It seems strange that a deficit in neuronal differentiation at E16.5 does not lead to changes in neuronal numbers until P7 (Figure 1). I would expect to see reduced neuronal numbers by P0 if there were differentiation defects at E16.5. What is the explanation for this discrepancy?

Response: Thanks for the reviewer's comments. As the reviewer anticipated, Figures 1F, G, and H demonstrate that significant differences in the number of neurons have already started to emerge in the CA1 and DG regions during the P0 period. Based on the hippocampal development process, CA3 neurons remain in the multipolar cell accumulation zone for a longer duration compared to CA1 neurons, leading to the delayed development of CA3 pyramidal neurons relative to CA1. This could potentially explain why no difference in neuronal numbers was observed in the CA3 region at P0.

3. Comments: Figure 5 is interpreted as *Klf7* knockout causes reduced neurogenesis, but again cell death was not analyzed and remains a possible interpretation for the lower numbers of EdU+ cells or GFP+ cells (in utero electroporation) in the hippocampus at P7.

Response: Thank you for the reviewer's valuable comments. Apoptosis staining revealed no abnormal apoptotic activity in the *Emx1-Cre;Klf7^{F/F}* group during the embryonic period or early postnatal stages. For further details, please refer to the revised manuscript on page 12, lines 29-33, and Figures S2A and S2B.

4. Comments: In Figure 5, it is not clear whether the EdU or in utero electroporation initially labels the same number of cells in Control and *Klf7* knockout hippocampus. An acute time point would be necessary to determine whether the starting population is the same. This issue will also impact the interpretation of the results, especially IUE at P0, which occurs when there are already phenotypic differences observed in the hippocampus.

Response: We thank the reviewer for pointing this out. As shown in Figures 2D and 2G, there was no significant difference in the number of EdU+ cells between the control and *Emx1-Cre;Klf7^{F/F}* groups 4 hours after EdU labeling at E14.5. During the IUE procedure, identical experimental parameters (e.g., voltage and pulse duration) were used, and mice were randomly assigned to groups to minimize individual variability. As no developmental abnormalities were observed at E14.5, we performed the IUE experiment at P0 and collected samples to evaluate the initial population of EGFP+ cells. Typically, fluorescence becomes detectable approximately 24 hours post-transfection; therefore, samples were collected 36 hours after IUE for analysis. A representative image and quantification are provided below. No significant difference was found in the number of initial EGFP+ cells in the HNE between the two groups.

NOTE: We have removed unpublished data that had been provided for the referees in confidence.

5. Comments: Figure 6 and 7 show there are clearly defects in the connectivity of the hippocampal neurons in the *Klf7* knockout. One of the most striking findings is the complete lack of CA inputs from the contralateral side in the *Klf7* knockout. When I saw this result, I thought that maybe the contralateral input is necessary for proper synaptic/circuit integration and might be required for cell survival. Again, I wonder if the neurons are made more or less normally, but do not receive proper input and start to die or shrink postnatally.

Response: We thank the reviewer for their valuable comments. Based on the results of apoptosis staining, no significant abnormalities in cell death were observed in the hippocampus during development. Accordingly, we have revised the manuscript in Section 3.3 to include the following: "...we first evaluated apoptotic activity during development. Immunostaining for cleaved caspase-3 revealed no significant difference in programmed cell death between *Klf7*-deficient mice and *Klf7*^{F/F} controls at E14.5. Similarly, cleaved caspase-3-positive cells remained unchanged at E17.5, P0, and P7 (Figures S2A and S2B).". Please see the revised manuscript on page 12, lines 29-33.

6. Comments: Why was the P7 timepoint chosen for the RNA-seq experiment in Figure 9? By P7 the hippocampus phenotype of fewer neurons is quite significant. It seems like a better experiment would be to look a bit earlier (P0?) to see what is changing before the neuronal loss occurs.

Response: We sincerely appreciate the reviewer's valuable comments. In our initial sequencing, we used hippocampal tissue from P0-stage mice. However, due to the small tissue size at this stage, the results showed considerable variability between groups. To improve consistency and data reliability, we subsequently performed RNA sequencing using hippocampal tissue from P7-stage mice.

7. Comments: The results in Figure 9 after overexpression of Draxin are confusing. It appears that overexpressing Draxin in Control reduces the number of GFP+ cells, but overexpression in *Klf7* knockouts increases the number of GFP+ cells. This discrepancy in the effects of Draxin on the control vs. *Klf7* knockout are not explained. I think this is important to have strong data for this experiment because the title claims "Draxin-mediated neuronal migration".

Response: We sincerely thank the reviewer for pointing this out. In the Methods section, we stated: "A solution containing plasmid DNA (2.5 µg/µL or 1.5 µg/µL) and Fast Green dye (Sigma-Aldrich) was injected into the lateral ventricles of the embryos," but we inadvertently omitted the plasmid concentrations in the Results section. We have now revised Section 3.8 to include: "Plasmids expressing EGFP (2.5 µg/µL) or Draxin-EGFP (1.5 µg/µL) were electroporated into precursor cells of the VZ at P0." Please see the revised manuscript on page 17, lines 22-23.

Due to space constraints in the manuscript, we provide additional clarification here. In the Draxin-EGFP group, the reduced plasmid concentration led to fewer Draxin-EGFP+ cells in the DG region of *Klf7*^{F/F} control mice compared to EGFP+ cells. However, in *Emx1-Cre;Klf7*^{F/F} mice, Draxin-EGFP+ cell numbers in the DG were significantly higher than EGFP+ cells. These results demonstrate that even low-level Draxin overexpression can effectively rescue the DG neuronal migration defect caused by *Klf7* deletion (Figures 9I and 9J).

Similarly, in IUE experiments from E14.5 to P0, the lower concentration of Draxin-EGFP resulted in fewer labeled cells in the CA region of *Klf7*^{F/F} control mice, but an increased number in the DG region. This suggests that modest Draxin overexpression promotes premature neuronal migration to the DG during development, underscoring its key role in regulating DG-specific migration.

8. Comments: It is not clear what data in the manuscript support the conclusion that *Klf7* knockout results in neuronal migration defects. It is suggested that the decrease in differentiated cells in the CA and DG regions of the hippocampus (Figure 5) is due to impaired migration, but there is no evidence that these cells get "stuck" in the hippocampal neuroepithelium. Is there evidence that indeed these cells do not migrate and are stuck?

Response: We thank the reviewer for the insightful comments. EdU was administered at E12.5 or E14.5, and brains were collected at P7 for staining and analysis. Since EdU incorporates into DNA during active cell division, it labels progenitor cells dividing at the time of injection. By analyzing EdU+ cell distribution at P7, we can track the migration and positioning of these cells and their progeny within the hippocampus.

As shown in Figures 2D and 2G, no significant difference was observed between the control and *Emx1-Cre; Klf7*^{F/F} groups in EdU+ progenitor cells 4 hours after EdU labeling at E14.5, suggesting similar initial proliferation. However, by P7 (Figure 5), a marked reduction in EdU+ cells was detected in the cKO hippocampus, indicating that a subset of labeled cells and their descendants failed to migrate properly to their target hippocampal regions.

To further investigate this, we used in utero electroporation, which allows for precise temporal and spatial gene delivery to ventricular zone progenitors, enabling the tracking of neuronal migration. As shown in Figures 4L-4O, *Emx1-Cre; Klf7^{F/F}* mice displayed a striking absence of EGFP+ cells in the CA and DG regions, supporting the conclusion that *Klf7* deletion disrupts neuronal migration during hippocampal development. These migration defects may involve arrested migration or misdirection of cells to ectopic regions.

Thanks again for the reviewers' positive comments and valuable suggestions to improve the quality of our manuscript.

Second decision letter

MS ID#: dev.204718R1

MS TITLE: KLF7 orchestrates hippocampal development through neurogenesis and Draxin-mediated neuronal migration

AUTHORS: Yitong Liu, Wentong Hong, Yuyan Zhou, Ailing Zhang, Pifang Gong, Guibo Qi, Xuan Song, Zhenru Wang, Xuanming Shi, Congcong Qi and Song Qin

Dear Dr Qin,

I have now received all the referees reports on the above manuscript, and have reached a decision. The referees' comments are appended below, or you can access them online: please go to .

The overall evaluation is positive and we would like to publish a revised manuscript in *Development*, provided that the remaining referees' comments can be satisfactorily addressed. Please address the remaining concerns regarding methods, over-expression experiments and antibody validity. This may require additional experiments and/or better rationale.

Please attend to all of the reviewers' comments in your revised manuscript and detail them in your point-by-point response. If you do not agree with any of their criticisms or suggestions explain clearly why this is so. If it would be helpful, you are welcome to contact us to discuss your revision in greater detail. Please send us a point-by-point response indicating your plans for addressing the referees' comments, and we will look over this and provide further guidance.

Reviewer 1

Advance summary and potential significance to field

The authors have addressed all of my previous comments, including several through additional experiments. The revisions improve the clarity of the manuscript and further strengthen the overall conclusions. However, two issues still warrant further attention:

1. Breeding scheme to generate the knockout mice

The authors clarified that *Emx1-Cre; Klf7^{F/F}* mice were used as breeders to generate conditional knockout animals. In the revised manuscript, they state: "*Emx1-Cre; Klf7^{F/F}* mice were mated with *Klf7^{F/F}* mice of different genotypes (*Emx1-Cre; Klf7^{F/F}* and *Klf7^{F/F}*". This wording remains unclear. Do the authors mean that *Emx1-Cre; Klf7^{F/F}* mice were mated with *Klf7^{F/F}* mice, or were some breedings between two *Emx1-Cre; Klf7^{F/F}* mice? The use of conditional knockout animals with known behavioral phenotypes as breeders raises a major concern, as it could significantly affect offspring brain development and behavior. This is an unusual breeding strategy and warrants further clarification. Specifically, the authors should:

1. Clearly state the genotypes of the male and female breeders in the Methods, and
2. Include a dedicated paragraph in the Discussion addressing the potential caveats of this breeding scheme.

2. Prox1 immunofluorescence data (Fig. 9I)

The new data provided using Prox1 immunofluorescence is not convincing. It is difficult to determine whether PROX1 signals truly co-localize with GFP. In addition, some PROX1 signals do not appear to be nuclear, which raises concerns about antibody specificity or staining quality.

Minor comments:

* Page 12, lines 35-36: The sentence "and undergo rapid proliferation (approximately 8-18 hours) and undergo active proliferation" is repetitive and should be revised for clarity.

* Page 10, line 35: The sentence referencing Emx1-Cre should cite the original source: Gorski et al., 2002.

Reviewer 2

Advance summary and potential significance to field

Comments for the author

Referring to this exchange:

7. Comments: The results in Figure 9 after overexpression of Draxin are confusing. It appears that overexpressing Draxin in Control reduces the number of GFP+ cells, but overexpression in Klf7 knockouts increases the number of GFP+ cells. This discrepancy in the effects of Draxin on the control vs. Klf7 knockout are not explained. I think this is important to have strong data for this experiment because the title claims "Draxin-mediated neuronal migration".

Response: We sincerely thank the reviewer for pointing this out. In the Methods section, we stated: "A solution containing plasmid DNA (2.5µg/µL or 1.5µg/µL) and Fast Green dye (Sigma-Aldrich) was injected into the lateral ventricles of the embryos,"but we inadvertently omitted the plasmid concentrations in the Results section. We have now revised Section 3.8 to include: "Plasmids expressing EGFP (2.5µg/µL) or Draxin-EGFP (1.5µg/µL) were electroporated into precursor cells of the VZ at P0." Please see the revised manuscript on page 17, lines 22-23. Due to space constraints in the manuscript, we provide additional clarification here. In the Draxin-EGFP group, the reduced plasmid concentration led to fewer Draxin-EGFP+ cells in the DG region of Klf7F/F control mice compared to EGFP+ cells. However, in Emx1-Cre;Klf7F/F mice, Draxin-EGFP+ cell numbers in the DG were significantly higher than EGFP+ cells. These results demonstrate that even low-level Draxin overexpression can effectively rescue the DG neuronal migration defect caused by Klf7 deletion (Figures 9I and 9J). Similarly, in IUE experiments from E14.5 to P0, the lower concentration of Draxin-EGFP resulted in fewer labeled cells in the CA region of Klf7F/F control mice, but an increased number in the DG region. This suggests that modest Draxin overexpression promotes premature neuronal migration to the DG during development, underscoring its key role in regulating DG-specific migration.

Why were different concentrations of plasmid DNA used for EGFP and Draxin-EGFP electroporations? There is no explanation and it makes the results more difficult to interpret and for the reader to understand. Also, the difference in plasmid concentration is only <2 fold, but the difference in cell counts (EGFP-Cntrl vs. Draxin;EGFP-Cntrl) is >50 reduction in DG (Figure 9J) and ~80% reduction in CA1. Please let me know if I am missing something here, but it seems that overexpressing Draxin has a different effect in Controls than replacing the lost Draxin in the knockout. This should at the least be addressed in the Results section to avoid reader confusion and explain this dissonance of results.

Second revision

Author response to reviewers' comments

Point-by-point responses to Reviewers' comments

Comments from the Reviewers:

Reviewer 1: The authors have addressed all of my previous comments, including several through additional experiments. The revisions improve the clarity of the manuscript and further strengthen the overall conclusions. However, two issues still warrant further attention:

1. Breeding scheme to generate the knockout mice

The authors clarified that $Emx1-Cre; Klf7^{F/F}$ mice were used as breeders to generate conditional knockout animals. In the revised manuscript, they state: " $Emx1-Cre; Klf7^{F/F}$ mice were mated with $Klf7^{F/F}$ mice of different genotypes ($Emx1-Cre; Klf7^{F/F}$ and $Klf7^{F/F}$ ". This wording remains unclear. Do the authors mean that $Emx1-Cre; Klf7^{F/F}$ mice were mated with $Klf7^{F/F}$ mice, or were some breedings between two $Emx1-Cre; Klf7^{F/F}$ mice? The use of conditional knockout animals with known behavioral phenotypes as breeders raises a major concern, as it could significantly affect offspring brain development and behavior. This is an unusual breeding strategy and warrants further clarification. Specifically, the authors should:

1. Clearly state the genotypes of the male and female breeders in the Methods, and
2. Include a dedicated paragraph in the Discussion addressing the potential caveats of this breeding scheme.

Response: We thank the reviewer for their insightful comments. We have revised the Methods section to explicitly state the breeding scheme: " $Emx1-Cre; Klf7^{F/F}$ males were crossed with $Klf7^{F/F}$ females to generate experimental animals. No breeding pairs consisted of two $Emx1-Cre; Klf7^{F/F}$ mice." (please see revised Methods on page 4, lines 22-24).

Additionally, we have added the following paragraph to the Discussion to address potential caveats of this strategy: "To generate conditional knockout mice, male $Emx1-Cre; Klf7^{F/F}$ mice were mated with female $Klf7^{F/F}$ mice. This strategy efficiently yields experimental mutant ($Emx1-Cre; Klf7^{F/F}$) and control ($Klf7^{F/F}$) littermates. We acknowledge that using mutant males as breeders could potentially introduce subtle effects on offspring development or behavior. To rigorously control for this possibility and all background variables, all experimental comparisons, including behavioral analyses, were conducted exclusively between littermates. While no overt phenotypic differences were observed in breeders, the potential contribution of this strategy to the observed phenotypes, particularly behavior, cannot be entirely excluded." Please see revised Discussion on page 20, lines 15-23.

2. *Prox1* immunofluorescence data (Fig. 9I)

*The new data provided using *Prox1* immunofluorescence is not convincing. It is difficult to determine whether PROX1 signals truly co-localize with GFP. In addition, some PROX1 signals do not appear to be nuclear, which raises concerns about antibody specificity or staining quality.*

Response: We sincerely thank the reviewer for the insightful and constructive comments. The original images in Figure 9I were acquired using an EVOS microscope, which is particularly well-suited for capturing the overall morphology of EGFP⁺ neurons, including their protrusions. However, this imaging approach can occasionally cause overexposure of the cell bodies, potentially leading to the impression that PROX1 and GFP are not co-localized.

To address this concern and provide clearer evidence, we have re-imaged the relevant sections using a confocal microscope, which offers superior optical sectioning and more precise signal localization. The revised images now more clearly demonstrate the nuclear co-localization of PROX1 and GFP.

Regarding the observation of non-nuclear PROX1 signals, we used the PROX1 monoclonal antibody from Proteintech (Cat. No. 67438-1-Ig). In our staining, this antibody primarily labels neuronal nuclei, consistent with PROX1's known nuclear function. However, we also observed faint yet detectable immunoreactivity in the protrusions of granule neurons, which may reflect low-level cytoplasmic expression or subcellular localization deserving further investigation. Notably, this antibody has been successfully used in immunofluorescence applications in peer-reviewed studies (e.g., <https://www.sciencedirect.com/science/article/pii/S1534580724000029>), supporting its specificity and reliability.

Minor comments:

Page 12, lines 35-36: The sentence "and undergo rapid proliferation (approximately 8-18 hours) and undergo active proliferation" is repetitive and should be revised for clarity.

Response: We sincerely thank the reviewer for careful reading. We have checked the manuscript and revised the description of "NPCs display a short cell cycle (~8-18 hours) and undergo robust proliferation." Please see page 12, lines 36-37.

Page 10, line 35: The sentence referencing *Emx1-Cre* should cite the original source: Gorski et al., 2002.

Response: We sincerely thank the reviewer for pointing this out. We have now added the original reference for *Emx1-Cre* as suggested (Gorski et al., 2002) in the revised manuscript on page 11, line 1.

Reviewer 2: SUMMARY OF THE ADVANCE MADE IN THIS PAPER AND ITS POTENTIAL SIGNIFICANCE TO THE FIELD

SUGGESTIONS TO AUTHORS

Referring to this exchange:

7. Comments: The results in Figure 9 after overexpression of *Draxin* are confusing. It appears that overexpressing *Draxin* in Control reduces the number of GFP+ cells, but overexpression in *Klf7* knockouts increases the number of GFP+ cells. This discrepancy in the effects of *Draxin* on the control vs. *Klf7* knockout are not explained. I think this is important to have strong data for this experiment because the title claims "*Draxin*-mediated neuronal migration".

Response: We sincerely thank the reviewer for pointing this out. In the Methods section, we stated: "A solution containing plasmid DNA (2.5 µg/µL or 1.5 µg/µL) and Fast Green dye (Sigma-Aldrich) was injected into the lateral ventricles of the embryos," but we inadvertently omitted the plasmid concentrations in the Results section. We have now revised Section 3.8 to include: "Plasmids expressing EGFP (2.5 µg/µL) or *Draxin*-EGFP (1.5 µg/µL) were electroporated into precursor cells of the VZ at P0." Please see the revised manuscript on page 17, lines 22-23. Due to space constraints in the manuscript, we provide additional clarification here. In the *Draxin*-EGFP group, the reduced plasmid concentration led to fewer *Draxin*-EGFP+ cells in the DG region of *Klf7^{F/F}* control mice compared to EGFP+ cells. However, in *Emx1-Cre; Klf7^{F/F}* mice, *Draxin*-EGFP+ cell numbers in the DG were significantly higher than EGFP+ cells. These results demonstrate that even low-level *Draxin* overexpression can effectively rescue the DG neuronal migration defect caused by *Klf7* deletion (Figures 9I and 9J). Similarly, in IUE experiments from E14.5 to P0, the lower concentration of *Draxin*-EGFP resulted in fewer labeled cells in the CA region of *Klf7^{F/F}* control mice, but an increased number in the DG region. This suggests that modest *Draxin* overexpression promotes premature neuronal migration to the DG during development, underscoring its key role in regulating DG-specific migration.

Why were different concentrations of plasmid DNA used for EGFP and *Draxin*-EGFP electroporations? There is no explanation and it makes the results more difficult to interpret and for the reader to understand. Also, the difference in plasmid concentration is only <2 fold, but the difference in cell counts (EGFP-Cntrl vs. *Draxin*;EGFP-Cntrl) is >50 reduction in DG (Figure 9J) and ~80% reduction in CA1. Please let me know if I am missing something here, but it seems that overexpressing *Draxin* has a different effect in Controls than replacing the lost *Draxin* in the knockout. This should at the least be addressed in the Results section to avoid reader confusion and explain this dissonance of results.

Response: We sincerely appreciate the reviewer's insightful comments regarding the use of different plasmid concentrations in our electroporation experiments (EGFP: 2.5 µg/µL vs. *Draxin*-EGFP: 1.5 µg/µL) and the interpretation of these results. We fully acknowledge that this concentration difference introduces complexity to data interpretation and apologize for not clearly explaining the rationale in the original manuscript. We now provide the following detailed clarification:

Rationale for Using Different Plasmid Concentrations:

The primary aim of the rescue experiment (*Draxin* overexpression in *Emx1-Cre; Klf7^{F/F}* mice) was to rigorously demonstrate that the observed rescue of migration defects was specifically due to *Draxin* function, rather than potential artifacts from high-level non-specific overexpression. To preclude this confounding factor and strengthen the validity of our conclusions, we intentionally used a

lower concentration (1.5 µg/µL) for the Draxin-EGFP plasmid compared to the control EGFP plasmid (2.5 µg/µL).

As seen in the raw images of Figure 9 and the quantification in Figure 9J, the lower concentration resulted in visibly reduced fluorescence intensity in the VZ and a significantly lower number of Draxin-EGFP⁺ cells in the DG of control mice. Despite this, Draxin overexpression still significantly increased the number of GFP⁺ cells in the DG of *Emx1-Cre; Klf7^{F/F}* mice, clearly indicating that the rescue effect is attributable to Draxin itself and not to non-specific expression levels. This design choice enhances the robustness and specificity of the rescue evidence.

Interpretation of Results in Control (*Klf7^{F/F}*) Mice:

DG Region (~50% Reduction): In *Klf7^{F/F}* control mice, electroporation with the lower-concentration Draxin-EGFP plasmid led to approximately a 50% reduction in the number of GFP⁺ cells in the DG compared to the higher-concentration EGFP control. This ~2-fold reduction aligns reasonably with the difference in plasmid concentration, though the biological impact may be somewhat amplified.

CA1 Region (~80% Reduction): CA1 Region (~80% reduction): The reviewer's observation of an ~80% reduction in GFP⁺ cells in the CA region is well taken. While we currently lack a complete mechanistic explanation, we believe this result is consistent with the known function of Draxin. In the control background, even moderate Draxin overexpression appears to promote premature neuronal migration toward the DG, as previously noted in our results. This shift may deplete the CA region of labeled cells and reflects Draxin's ability to direct migratory behavior toward DG-specific trajectories. Thus, both the reduction in CA1 labeling and the rescue phenotype in KLF7 cKO mice reflect the same biological principle: Draxin promotes DG-directed migration.

Acknowledgment of Experimental Complexity and Clarification in Manuscript:

We acknowledge that the concentration difference may have caused confusion, and we apologize for not clarifying this point earlier. To address this, we have revised Section 3.8 of the Results to include the rationale:

“To confirm the role of Draxin as a downstream effector in KLF7-regulated neuronal migration and preclude non-specific overexpression artifacts, we electroporated precursor cells of the VZ at P0 with either EGFP (2.5 µg/µL; control) or Draxin-EGFP (1.5 µg/µL) plasmids.” “While Draxin overexpression failed to rescue ..., Draxin-EGFP⁺ cells were ectopically located in the DG region of *Emx1-Cre;Klf7^{F/F}* mice (Figure 9L). These results demonstrate that Draxin selectively promotes neuronal migration to the DG and plays a critical role in region-specific hippocampal development.” (Please see page 17, lines 24-26 and 32-36.)

Justification for Not Repeating the Rescue Experiment:

While we carefully considered re-running the rescue experiment at matched plasmid concentrations, we ultimately decided against it due to the extensive technical demands and ethical considerations. Each in utero electroporation targeting the hippocampal primordium, followed by postnatal survival, processing, staining, imaging, and quantification, requires a minimum of 60 days per animal. The original rescue series took over six months to complete. In alignment with the 3Rs principle—particularly **Reduction**—we believe that repeating the experiment would not be ethically justified, as the current data robustly support our central conclusion.

We hope this detailed explanation and the corresponding manuscript revisions adequately address the reviewer's concerns. We are grateful for the opportunity to clarify this point.

Third decision letter

MS ID#: dev.204718R2

MS TITLE: KLF7 orchestrates hippocampal development through neurogenesis and Draxin-mediated neuronal migration

AUTHORS: Yitong Liu, Wentong Hong, Yuyan Zhou, Ailing Zhang, Pifang Gong, Guibo Qi, Xuan Song, Zhenru Wang, Xuanming Shi, Congcong Qi and Song Qin

Dear Dr Qin,

I am happy to tell you that your manuscript has been accepted for publication in Development, pending our standard publication integrity checks.